# Iterative Teaching by Label Synthesis

**Weiyang Liu[1,2,*]   Zhen Liu[3,*]   Hanchen Wang[1,*]   Liam Paull[3,4]   Bernhard Schölkopf[2]   Adrian Weller[1,5]**

[1]University of Cambridge   [2]MPI for Intelligent Systems, Tübingen   [3]Mila, Université de Montréal   [4]CIFAR AI Chair
[5]The Alan Turing Institute   [*]Equal Contribution   ✉{wl396@cam.ac.uk, zhen.liu.2@umontreal.ca}

## Abstract

In this paper, we consider the problem of iterative machine teaching, where a teacher provides examples sequentially based on the current iterative learner. In contrast to previous methods that have to scan over the entire pool and select teaching examples from it in each iteration, we propose a label synthesis teaching framework where the teacher randomly selects input teaching examples (*e.g.*, images) and then synthesizes suitable outputs (*e.g.*, labels) for them. We show that this framework can avoid costly example selection while still provably achieving exponential teachability. We propose multiple novel teaching algorithms in this framework. Finally, we empirically demonstrate the value of our framework.

## 1   Introduction

Machine teaching [103, 106] studies the problem of constructing a minimal dataset for a target concept such that a learner can learn the concept based on this dataset. Machine teaching has diverse applications ranging from crowd sourcing [71, 72, 100, 101] to model robustness [2, 3, 54, 63]. Machine teaching also has nice connections with curriculum learning [7] and coresets [1, 28].

Based on the learner, machine teaching can be performed in either batch or sequential fashion. The majority of prior work studies batch machine teaching [42, 56, 102, 103], where a teacher constructs a minimal batch set of training samples and provides it to a student in one shot without further interactions. Then the student keeps learning from this batch dataset for the target concept. The size of such a minimal batch set is called *teaching dimension* [22]. Differently, sequential machine teaching [38, 43, 44, 61] bridges the gap between machine teaching and practical learning algorithms by studying the sequential (*i.e.*, itera-

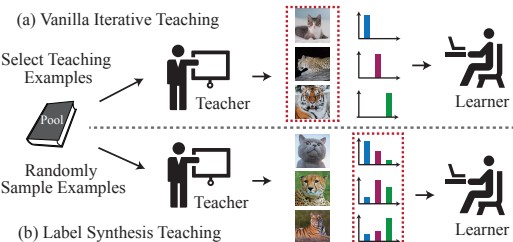

Figure 1: Comparison of vanilla iterative machine teaching and label synthesis teaching. The red dotted frames indicate the teacher's efforts.

tive) learner such as neural networks. A typical example is iterative machine teaching (IMT) [43, 44] where the teacher guides a learner to a target concept by interacting with the learner (*e.g.*, feeding training samples) in every iteration. The minimum number of such iterations is called the *iterative teaching dimension*. One of the largest challenges in sequential teaching is how to effectively and efficiently provide teaching examples to the iterative learner. Usually we are mostly interested in the pool-based teaching in IMT since it well matches the setting of modern machine learning. However, exploring all the possible teaching trajectories is computationally prohibitive. For example, there are $\binom{m}{k}^n$ possible teaching trajectories ($n$ is the number of iterations) if we select $k$ samples per iteration from a pool of size $m$. Due to such a huge search space, the selection of teaching examples is a combinatorial problem that is inherently difficult to solve. IMT [43] performs the teaching sample selection with a greedy policy, but it could be substantially sub-optimal in certain cases [38] and its computational complexity also scales linearly with the size of the dataset.

35th Conference on Neural Information Processing Systems (NeurIPS 2021).

We propose a general teaching framework called LAbel Synthesis Teaching (LAST), which in its standard version avoids the problems of sample selection – though we will later discuss how sample selection can be combined with our approach. In the standard version of LAST, teaching examples are randomly sampled (similar to SGD) in the pool and the teacher synthesizes their labels in order to quickly guide the learner to the desired model. A brief comparison between IMT and LAST is given in Fig. 1. LAST restricts the teaching to the label space and bypasses the selection of teaching samples. Therefore, LAST avoids the high complexity of selecting teaching samples in a large pool.

Intuition for why LAST is able to achieve promising teaching performance comes from the empirical success of knowledge distillation [26] and label smoothing [73]. Knowledge distillation shows that training neural networks with soft labels from a pretrained model can generally improve generalization. Label smoothing demonstrates that the ground truth labels are not necessarily the optimal supervision for training a model. Instead, smoothing the label with uniform distribution can calibrate the model and lead to better generalization. Both methods can be viewed as providing an alternative label (instead of the ground truth) to the learner in order to improve its generalizability. Moreover, [77, 78] show that there exists privileged information beyond the ground truth labels that can significantly improve the convergence rate of learning algorithms. Therefore, now we can safely argue that the ground truth labels are not always optimal learning signals. Motivated by these work, we aim to construct a teacher model that can adaptively synthesize suitable labels for a learner (with the hope to implicitly encode priviledged information) in order to improve the learner's convergence.

Specifically, we study LAST primarily under the omniscient scenario where the teacher knows everything about the learner (*e.g.*, the optimal learner parameters). To perform omniscient teaching, we consider a greedy teacher and a parameterized teacher. We show that greedy teaching can achieve exponential teachability (ET) [43] without selecting teaching examples. Additionally, we touch upon the black-box teaching scenario where the teacher knows less about the learner (*e.g.*, the optimal learner parameters are unavailable), and discuss how to perform LAST in this case.

LAST provides a unified view for understanding soft label methods, *e.g.*, knowledge distillation [26, 53], label smoothing [11, 59, 73], and self-training [104]. All these methods can be interpreted as modifying the labels to achieve desirable learning behavior and outcome. With LAST, we can connect iterative machine teaching to many classic learning algorithms and shed novel light on them.

## 2  Related Work

**Machine teaching**. Batch machine teaching [42, 56, 102, 103] has drawn increasing more attention recently. Different learners typically exhibit distinct behaviors during teaching. [10, 74] focus on how to teach version space learners. The teaching dimension of linear learners [42], kernel learners [36] and reinforcement learner [95] is extensively studied. Teaching a forgetful learner is explored in [29, 44]. [94, 105] consider how to teach multiple learners simultaneously. Sequential machine teaching [38, 43, 44, 91] studies iterative learners by considering the specific optimization algorithm that the learner adopts. Unlike batch scenario, the teaching quality is evaluated by the learner's convergence. [38] connects sequential teaching to optimal control and gains interesting insights, but it can not produce a practical teaching policy. [91] uses locality-sensitive sampling to scale IMT to large-scale problems. Machine teaching is shown useful in reinforcement learning [24, 32, 63, 76], human-in-the-loop learning [9, 31, 55], crowd sourcing [72, 100, 101] and cyber security [2, 57, 96, 97, 98]. [9, 13, 21, 35, 65, 108] study machine teaching from a more theoretical point of view.

**Cooperative communication**. Cooperative communication [69] is a mutual theory of mind reasoning between a teacher and a learner. The teacher selects data to convey a hypothesis and the learner infers a hypothesis given the selected data. Both agents have the shared goal of successfully transmitting beliefs. [84] formulates cooperative communication with the framework of optimal transport. In this framework, pedagogical reasoning [66, 67, 68], cooperative inference [82, 83, 93] and Bayesian teaching [15, 16, 92] can all be viewed as special cases. Among this line of research, cooperative inference [82] shares many similarities to our work and also adopts an iterative teaching paradigm. While [82] formulates the teaching problem with a probabilistic inference framework, LAST addresses the iterative teaching with a nested optimization framework.

**Soft-supervised learning**. It has been shown that supervising the model with soft labels can be beneficial to generalization [73, 89] and robustness [75, 107]. Label smoothing [73] replaces one-hot label vector with a mixture of itself and the uniform distribution, achieving stronger generalization in training neural networks. Knowledge distillation [26] uses the soft predicted label from a pretrained model to supervise the training of a student model, which can make the student model generalize

better than a trained-from-scratch one. [89] perturbs the one-hot label vector with random noise and also observes better generalization. Large-margin softmax [45, 50, 51, 52] can be approximately viewed as training learners with dynamic soft labels (Appendix E). Self-training [8, 90, 104] also uses dynamic soft labels to iteratively train the learner. In constrast to the empirical nature of soft-supervised learning, LAST formulates it as a unified and principled iterative teaching framework.

# 3 Label Synthesis Teaching

**Teaching protocol**. We first consider the simplest teaching protocol following [43]. Both the teacher and the learner observe the same sample $\mathcal{A}$ and share the same feature space by representing $\mathcal{A}$ as $\boldsymbol{x}$ with the label $\boldsymbol{y}$. The teacher has access to all the information about the learner such as the model parameter $\boldsymbol{w}^i$ at the $i$-th iteration, the learning rate $\eta_i$, the loss function $\ell$ and the specific optimization algorithm (*e.g.*, SGD). For interaction, the teacher can only communicate with the learner via examples $\{(\boldsymbol{x}_j^i, \boldsymbol{y}_j^i)\}_{j=1,\cdots,m}$ in the $i$-th iteration, where $m$ denotes the batch size.

**Teaching objective**. The goal of the teacher is to provide examples in each iteration such that the learner parameters $\boldsymbol{w}$ converge to desired parameters $\boldsymbol{w}^*$ as fast as possible. For example, $\boldsymbol{w}^*$ typically is $\arg\min_{\boldsymbol{w}} \mathbb{E}_{(\boldsymbol{x},\boldsymbol{y})}\{\ell(\boldsymbol{x},\boldsymbol{y}|\boldsymbol{w})\}$. Therefore, a general objective for any teaching algorithm is $\min_{\{(\boldsymbol{x}^1,\boldsymbol{y}^1),\cdots,(\boldsymbol{x}^T,\boldsymbol{y}^T)\}} d(\boldsymbol{w}^T,\boldsymbol{w}^*)$ where $T$ denotes the termination iteration for the teaching algorithm. We use batch size 1 here, but it is straightforward to extend the batch size to arbitrary $m$. $d(\cdot,\cdot)$ denotes some discrepancy measure (*e.g.*, Euclidean distance). This objective aims to find a teaching trajectory that is terminated at the $T$-th step such that the discrepancy between the current learner parameters and the desired learner parameters is the smallest. This is very challenging given the enormous search space. To simplify the problem, [43] proposes a greedy policy to generate the teaching curriculum, but it is computationally expensive when dealing with a large pool.

## 3.1 The LAST Framework

Unlike the general teaching objective that optimizes the sequential sample selection, the LAST framework instead optimizes the teaching examples in the label space, and the corresponding objective becomes $\min_{\{\boldsymbol{y}^1,\cdots,\boldsymbol{y}^T\}} d(\boldsymbol{w}^T,\boldsymbol{w}^*)$ where $\boldsymbol{x}^1,\cdots,\boldsymbol{x}^T$ are sampled uniformly from the pool (same as SGD). Such a teaching objective is general and does not make any assumption on the optimal structure in the teaching trajectory, making the problem intractable. To simplify the problem, we can assume that the teaching signal (*i.e.*, label $\boldsymbol{y}$ in LAST) only depends on the ran-

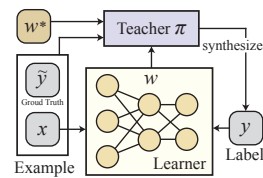

Figure 2: Overview.

domly sampled $\{\boldsymbol{x}, \tilde{\boldsymbol{y}}\}$ and the learner's current and desired model (*i.e.*, parameters). Therefore, we can formulate the corresponding label synthesis policy as $\boldsymbol{y} = \pi_{\boldsymbol{\theta}}(\boldsymbol{x}, \tilde{\boldsymbol{y}}, \boldsymbol{w}^t, \boldsymbol{w}^*)$ where $\pi$ is optionally parameterized by $\boldsymbol{\theta}$ and $\tilde{\boldsymbol{y}}$ denotes the original ground truth label. Then the teaching objective becomes $\arg\min_{\pi_{\boldsymbol{\theta}}} d(\boldsymbol{w}^T, \boldsymbol{w}^*)$. An overview of LAST is given in Fig. 2. Moreover, LAST provides a unified framework for knowledge distillation, label smoothing and self-training. Knowledge distillation [26] generates the label with the policy $\boldsymbol{y}_i = \frac{\exp(\boldsymbol{z}_i(\boldsymbol{x})/\psi)}{\sum_j \exp(\boldsymbol{z}_j(\boldsymbol{x})/\psi)}$ where $\psi$ is the temperature and $\boldsymbol{z}_i(\boldsymbol{x})$ is the $i$-th logit output of $\boldsymbol{x}$ from a pretrained neural network. Label smoothing [73] uses $\boldsymbol{y} = \mu\tilde{\boldsymbol{y}} + \frac{1-\mu}{K}\mathbf{1}$ ($K$ is the number of classes) as the label synthesis policy. Self-training [104] feeds the learner with the pseudo-label predicted by the learner from the last iteration. All these methods can be viewed as using some form of customized label synthesis policies.

## 3.2 Greedy Teaching Policy

For the $t$-th iteration, we can approximately solve the original teaching problem with the optimization $\min_{\{\boldsymbol{y}^{t+1},\cdots,\boldsymbol{y}^{t+v}\}} d(\boldsymbol{w}^{t+v}, \boldsymbol{w}^*)$ where $v \leq T$ is the number of steps being considered. By using $v = 1$ and the gradient descent update in LAST, we obtain a greedy policy $\arg\min_{\boldsymbol{y}^t} d(\boldsymbol{w}^{t+1}, \boldsymbol{w}^*)$ where $\boldsymbol{w}^{t+1}$ is one-step update from the current learner $\boldsymbol{w}^t$ *w.r.t.* the learner loss $\ell$ and example $\{\boldsymbol{x}^t, \boldsymbol{y}^t\}$. The greedy policy minimizes the Euclidean distance between the current learner parameters and the desired parameters in each iteration by learning a suitable label. We apply the greedy policy to each iteration for generating labels. The label for the example $\boldsymbol{x}^t$ at the $t$-th iteration is produced by

$$\min_{\boldsymbol{y}^t}\left\{\underbrace{\left\|\boldsymbol{w}^{t-1} - \eta_t\frac{\partial\ell(\boldsymbol{x}^t,\boldsymbol{y}^t|\boldsymbol{w}^{t-1})}{\partial\boldsymbol{w}^{t-1}} - \boldsymbol{w}^*\right\|_2^2}_{G(\boldsymbol{x}^t,\boldsymbol{y}^t|\boldsymbol{w}^{t-1})}\right\} \Rightarrow \min_{\boldsymbol{y}^t}\left\{\underbrace{\eta_t^2\left\|\frac{\partial\ell(\boldsymbol{x}^t,\boldsymbol{y}^t|\boldsymbol{w}^{t-1})}{\partial\boldsymbol{w}^{t-1}}\right\|_2^2}_{T_1(\boldsymbol{x}^t,\boldsymbol{y}^t|\boldsymbol{w}^{t-1})} - \underbrace{2\eta_t\langle\boldsymbol{w}^{t-1}-\boldsymbol{w}^*, \frac{\partial\ell(\boldsymbol{x}^t,\boldsymbol{y}^t|\boldsymbol{w}^{t-1})}{\partial\boldsymbol{w}^{t-1}}\rangle}_{T_2(\boldsymbol{x}^t,\boldsymbol{y}^t|\boldsymbol{w}^{t-1})}\right\} \quad (1)$$

where $\ell(\boldsymbol{x}^t, \boldsymbol{y}^t|\boldsymbol{w}^t)$ denotes the loss function of the learner and $\boldsymbol{w}^t$ is the learner parameter in the $t$-th iteration. For simplicity, we consider the mini-batch size as 1. It is straightforward to write Eq. (1)

in a mini-batch form. According to the decomposition of the parameter discrepancy, the teacher optimizes the labels such that the parameter discrepancy can be minimized. In contrast to IMT, which aims to optimize the sample $\boldsymbol{x}^t$, the greedy teaching in LAST first randomly samples the teaching example and then optimize the label based on $\arg\min_{\boldsymbol{y}^t} \eta_t^2 T_1(\boldsymbol{x}^t, \boldsymbol{y}^t | \boldsymbol{w}^t) - 2\eta_t T_2(\boldsymbol{x}^t, \boldsymbol{y}^t | \boldsymbol{w}^t)$. The overall procedure is given in Algorithm 1. Depending on the teacher's capability, we can optionally consider a few constraints for the synthesized labels.

**One-hot constraint**. For classification tasks, we can constrain the teacher to synthesize one-hot labels for $\boldsymbol{x}$. With the one-hot constraint, label synthesis becomes a standard integer programming problem [87].

---

**Algorithm 1** Omniscient Greedy LAST

---
Initialize $t = 1$, $\boldsymbol{w}^0$, $\epsilon$ and $T$;
**while** $\|\boldsymbol{w}^t - \boldsymbol{w}^*\|_2 \geq \epsilon$ or $t \leq T$ **do**
    Randomly select a sample $\boldsymbol{x}^t$ from the pool;
    Solve Eq. (1) to synthesize the label $\boldsymbol{y}^t$:
    Use the synthesized label $\boldsymbol{y}^t$ for the update:
$$\boldsymbol{w}^t = \boldsymbol{w}^{t-1} - \eta_t \frac{\partial \ell(\boldsymbol{x}^t, \boldsymbol{y}^t | \boldsymbol{w}^t)}{\partial \boldsymbol{w}^t};$$
    Set $t \leftarrow t + 1$;
**end**

---

**Soft constraint**. A straightforward relaxation of the one-hot constraint is to constrain the labels on a probability simplex, *i.e.*, $\sum_i \boldsymbol{y}_i = 1$ and $\boldsymbol{y}_i \geq 0, \forall i$, and it becomes a constrained optimization problem.

**Magnitude constraint**. For regression, we can constrain the $p$-norm of the label vector to be smaller than a positive constant $r$, *i.e.*, $\|\boldsymbol{y} - \langle \boldsymbol{w}^t, \boldsymbol{x}\rangle\|_p \leq r$ (for constraining the synthesized label to be around the current prediction) or $\|\boldsymbol{y} - \tilde{\boldsymbol{y}}\|_p \leq r$ (for constraining the synthesized label to be around the ground truth label), where we usually use $p = 2$. A larger magnitude $r$ indicates a more flexible and powerful teacher. For classification, we can use the same magnitude constraints (optionally) along with soft constraints $\sum_i \boldsymbol{y}_i = 1$ and $\boldsymbol{y}_i \geq 0, \forall i$.

**No constraint**. The most powerful teacher has no constraint on the synthesized labels and can synthesize any label vector, making it easy to solve the unconstrained optimization in Eq. (1).

### 3.2.1 Teaching Linear Learners

We start by discussing how to use greedy LAST to teach representative linear learners such as linear least square regression (LSR) and logistic regression (LR), and then give theoretical analyses on their iterative teaching dimension (*i.e.*, convergence properties). Specifically, we consider the following standard objective function: $\ell_{\text{LSR}}(\boldsymbol{x}, y | \boldsymbol{w}) = \frac{1}{2}(\langle \boldsymbol{w}, \boldsymbol{x}\rangle - y)^2$ for the LSR learner and $\ell_{\text{LR}}(\boldsymbol{x}, y | \boldsymbol{w}) = \log(1 + \exp\{-y\langle \boldsymbol{w}, \boldsymbol{x}\rangle\})$ for the LR learner. $\{\boldsymbol{x}_{[i]}, y_{[i]}\}$ is the $i$-th sample in the pool. We gain intuitions by looking at the synthesized label. Taking LSR as an concrete example, Eq (1) becomes

$$\min_y \eta_t^2 \langle \boldsymbol{x}, \boldsymbol{x}\rangle y^2 + \left(2\eta_t \langle \boldsymbol{w} - \boldsymbol{w}^*, \boldsymbol{x}\rangle - 2\eta_t^2 \langle \boldsymbol{x}, \boldsymbol{x}\rangle \langle \boldsymbol{w}, \boldsymbol{x}\rangle\right)y + \eta_t^2 \langle \boldsymbol{x}, \boldsymbol{x}\rangle \langle \boldsymbol{w}, \boldsymbol{x}\rangle - 2\eta_t \langle \boldsymbol{w}, \boldsymbol{x}\rangle \langle \boldsymbol{w} - \boldsymbol{w}^*, \boldsymbol{x}\rangle \quad (2)$$

from which we can obtain the closed-form solution for the synthesized label as

$$y^* = \underbrace{\frac{2\eta_t^2 \langle \boldsymbol{x}, \boldsymbol{x}\rangle \langle \boldsymbol{w}, \boldsymbol{x}\rangle - 2\eta_t \langle \boldsymbol{w} - \boldsymbol{w}^*, \boldsymbol{x}\rangle}{2\eta_t^2 \langle \boldsymbol{x}, \boldsymbol{x}\rangle}}_{\text{Synthesized label}} = (1 - \lambda) \cdot \underbrace{\langle \boldsymbol{w}, \boldsymbol{x}\rangle}_{\text{Predicted label -- easy}} + \lambda \cdot \underbrace{\langle \boldsymbol{w}^*, \boldsymbol{x}\rangle}_{\text{Optimal label -- hard}} \quad (3)$$

where $\lambda = (\eta_t \langle \boldsymbol{x}, \boldsymbol{x}\rangle)^{-1}$. $\lambda$ is essentially a parameter that weights the importance between using the current prediction as the training label and using the (pseudo) ground truth label as the training label. Suppose that the input data lies on a hypersphere with radius $\gamma$, then we have $\lambda = \frac{1}{\eta_t \gamma^2}$. Typically $\eta_t$ will gradually decrease during training, so $\lambda$ will gradually increase. Interestingly, this implies that the training label should be gradually shifted from the current predicted label to the ground truth label during training. This result validates our argument that it is not always optimal to feed the learner with ground truth labels. The predicted label is trivial to learn for the current learner (since the learner can output this same prediction without any update) and the optimal label is the most difficult to learn, revealing a principle that a learner should be taught in an easy-to-hard fashion in order to learn a concept efficiently. This implication nicely matches the conclusions drawn in curriculum learning [7], self-paced learning [37], cognitive shaping [17, 34, 62], continuation methods [4] and IMT [43, 44].

Moreover, greedy LAST has the property of teaching monotonicity [43] if the learner loss satisfies certain conditions (See Proposition 1 for a formal argument). This nice property implies that LAST can always converge to $\boldsymbol{w}^*$ faster than a random teacher (*i.e.*, SGD) if both of them sample teaching examples uniformly from the pool. An illustration of why LAST yields better convergence to $\boldsymbol{w}^*$ is given in Fig. 3. We can observe that LAST is guaranteed to better minimize the discrepancy between $\boldsymbol{w}^1$ and $\boldsymbol{w}^*$ for the first iteration if both LAST and SGD use the same initialization and training examples. Such a guarantee leads to the teaching monotonicity

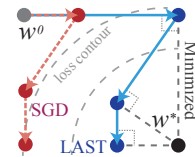

Figure 3: Example.

under a few conditions. If the learner loss is properly designed (*e.g.*, least square loss [43]) such that teaching monotonicity can be satisfied, then LAST will always converge faster than SGD.

**Proposition 1** (Teaching Monotonicity). *Given a learner $\ell$ and a randomly sampled teaching data point $(\boldsymbol{x}, \tilde{y}) \sim P$ for both LAST and a random teacher (SGD), if $\mathbb{E}\{G(\boldsymbol{x}, y|\boldsymbol{w}_1)\} \leq \mathbb{E}\{G(\boldsymbol{x}, y|\boldsymbol{w}_2)\}$ holds for any $\boldsymbol{w}_1, \boldsymbol{w}_2$ that satisfy $\mathbb{E}\{\|\boldsymbol{w}_1 - \boldsymbol{w}^*\|\} \leq \mathbb{E}\{\|\boldsymbol{w}_2 - \boldsymbol{w}^*\|\}$, then with the same initialization and learning rate, in expectation LAST can converge not slower than the random teacher.*

### 3.2.2 Teaching Neural Learners

We extend the greedy teaching to neural learners. As an illustrative example, we teach a two-layer multi-layer perceptron (MLP): $\ell_{\text{MLP}} = \text{CrossEntropy}(\text{SoftMax}(\boldsymbol{W}^\top \sigma(\boldsymbol{V}^\top \boldsymbol{x})), \boldsymbol{y})$ where $\boldsymbol{V} \in \mathbb{R}^{d_1 \times d_2}$ is the weight matrix of the first layer, $\boldsymbol{W} \in \mathbb{R}^{d_2 \times K}$ is the weight matrix of the output layer, $\sigma(\cdot)$ denotes an element-wise nonlinear function (*e.g.*, ReLU), CrossEntropy$(\cdot, \cdot)$ is the cross-entropy loss and SoftMax$(\cdot)$ is the softmax function. We denote the intermediate outputs as $\boldsymbol{U} = \boldsymbol{V}^\top \boldsymbol{x}$ and $\boldsymbol{P} = \sigma(\boldsymbol{U})$. Because the weights for the MLP are hierarchically structured, we propose a simple yet effective heuristic to teach MLP learners. The training label is generated by

$$\min_{\boldsymbol{y}} \underbrace{\left\| \boldsymbol{W}^t - \eta_t \frac{\partial \ell(\boldsymbol{x}, \boldsymbol{y}|\boldsymbol{V}^t, \boldsymbol{W}^t)}{\partial \boldsymbol{W}^t} - \boldsymbol{W}^* \right\|_F^2}_{\text{Discrepancy of the output layer: } \|\boldsymbol{W}^{t+1} - \boldsymbol{W}^*\|} + \beta \underbrace{\left\| \boldsymbol{V}^t - \eta_t \frac{\partial \ell(\boldsymbol{x}, \boldsymbol{y}|\boldsymbol{V}^t, \boldsymbol{W}^t)}{\partial \boldsymbol{P}^t} \circ \frac{\partial \boldsymbol{P}^t}{\partial \boldsymbol{U}^t} \circ \frac{\partial \boldsymbol{U}^t}{\partial \boldsymbol{V}^t} - \boldsymbol{V}^* \right\|_F^2}_{\text{Discrepancy of the first layer: } \|\boldsymbol{V}^{t+1} - \boldsymbol{V}^*\|} \quad (4)$$

where we are using the chain rule to compute $\frac{\partial \ell}{\partial \boldsymbol{V}^t}$. Both $\boldsymbol{P}^t$ and $\boldsymbol{U}^t$ are intermediate outputs at the $t$-th iteration. $\beta$ is a hyperparameter that balances the importance of weights in different layers. When $\beta = 1$, it becomes $\min_{\boldsymbol{y}} \|[\boldsymbol{W}^{t+1}; \boldsymbol{V}^{t+1}] - [\boldsymbol{W}^*; \boldsymbol{V}^*]\|_F^2$ which is equivalent to minimizing the Euclidean distance between the concatenated weight matrices and the desired ones.

## 3.3 Learning a Parameterized Teaching Policy

The greedy policy only considers $v = 1$ in $\min_{\{\boldsymbol{y}^{t+1}, \cdots, \boldsymbol{y}^{t+v}\}} d(\boldsymbol{w}^{t+v}, \boldsymbol{w}^*)$ and it is by no means optimal. We take a step further by considering $v > 1$. To approximate $\min_{\boldsymbol{y}^{t+1}, \cdots, \boldsymbol{y}^{t+v}} \|\boldsymbol{w}^{t+v} - \boldsymbol{w}^*\|_2^2$, we propose to learn a parameterized teaching policy $\pi_{\boldsymbol{\theta}}$ through $\min_{\pi(\boldsymbol{\theta})} \|\boldsymbol{w}^{t+v} - \boldsymbol{w}^*\|_2^2$.

**Unrolling**. We can unroll the optimization algorithm of the learner into the teaching objective. In general, we aim to solve the following bi-level nested optimization objective for the teacher:

$$\min_{\boldsymbol{\theta}} \|\boldsymbol{w}^v(\boldsymbol{\theta}) - \boldsymbol{w}^*\|_2^2 \quad \text{s.t.} \quad \boldsymbol{w}^v(\boldsymbol{\theta}) = \arg \min_{\boldsymbol{w}} \mathbb{E}_{\{\boldsymbol{x}, \tilde{\boldsymbol{y}}\}} \{\ell(\boldsymbol{x}, \pi_{\boldsymbol{\theta}}(\boldsymbol{x}, \tilde{\boldsymbol{y}}, \boldsymbol{w}^t, \boldsymbol{w}^*)|\boldsymbol{w})\} \quad (5)$$

where $\ell$ is the learner loss and $\pi_{\boldsymbol{\theta}}$ is the parameterized teaching policy that generates the label. For gradient descent learners, $\boldsymbol{w}^v(\boldsymbol{\theta})$ is obtained by unrolling $v$ steps of stochastic gradient descent *w.r.t.* $\ell$. Unrolling is shown useful in [12, 41, 46, 47, 58] and shares similar spirits with back-propagation through time in recurrent networks [85] and meta-learning [20]. The greedy policy is the optimal solution to Eq. (5) for $v = 1$, and the parameterized policy aims to approximate the solution to $v > 1$. For large $v$, unrolling builds a large computational graph for back-propagation and is very costly.

Optimizing the unrolled teaching policy is conceptually similar to optimizing recurrent neural networks with back-propagation through time, as illustrated in Fig. 4. Unrolling $v$ steps is equivalent to seeking a teaching policy that best minimizes the weight discrepancy after $v$-step gradient descent. Broadly speaking, our method also has intrinsic connections to learning an optimizer [5, 39] and teaching a loss function [88] in the sense that both aim to learn better gradients.

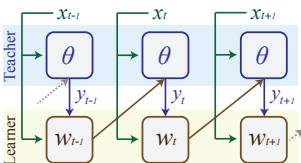

Figure 4: Unrolling the teacher.

**Policy gradient**. We can define $r_t = -\|\boldsymbol{w}^t - \boldsymbol{w}^*\|$ as the reward signal and directly apply policy gradient [86] to maximize it. The final teaching objective is to maximize the expected reward: $\max_{\boldsymbol{\theta}} J(\boldsymbol{\theta}) := \mathbb{E}_{(s_t, a_t) \sim \pi_{\boldsymbol{\theta}}(a|s)} \{\sum_{t=1}^T \gamma^t r_t(s_t, a_t)\}$ where $\gamma \in (0, 1]$ is the discounting factor, $T$ is the termination iteration and $(s_t, a_t)$ is the state-action pair at the $t$-th iteration. The state is characterized by the sample $\{\boldsymbol{x}, \tilde{\boldsymbol{y}}\}$, current learner $\boldsymbol{w}$ and optimal learner $\boldsymbol{w}^*$. The action space is a discretized label space. The policy gradient for updating $\boldsymbol{\theta}$ is given by $\nabla_{\boldsymbol{\theta}} J(\boldsymbol{\theta}) = \mathbb{E}_{\pi_{\boldsymbol{\theta}}(a_t|s_t)} \{R \nabla_{\boldsymbol{\theta}} \sum_{t=1}^T \log \pi_{\boldsymbol{\theta}}(a_t|s_t)\}$ where $R = \sum_{t=1}^T \gamma^t r_t(s_t, a_t)$. Appendix G gives the details for the state and action. Note that we stick to the simplest policy gradient for demonstration.

**Parameterization of the teacher**. The teaching policy can be parameterized as any neural network, such as MLP, CNN [25, 27, 33, 49, 52, 70] and transformer [14, 79]. It may also be beneficial to use dynamic networks [23, 30, 48], since the optimal teacher parameters for different input may vary.

## 3.4 Theoretical Guarantees and Justifications

We first study how label synthesis affects the gradients when the label is a scalar. For LSR learners, the gradient *w.r.t.* $\boldsymbol{w}$ of a single sample $(\boldsymbol{x}, \tilde{y})$ is $\nabla_{\boldsymbol{w}} \ell = (\langle \boldsymbol{w}, \boldsymbol{x} \rangle - \tilde{y})\boldsymbol{x}$. For LR learners, the gradient is

$\nabla_{\boldsymbol{w}}\ell = \frac{-\tilde{y}\boldsymbol{x}}{1+\exp(\tilde{y}\langle\boldsymbol{w},\boldsymbol{x}\rangle)}$. Modifying the label is essentially to re-scale the gradient without changing the direction. We denote the new gradient after modifying the label to $y$ as $\nabla'_{\boldsymbol{w}}\ell = g(y)\nabla_{\boldsymbol{w}}\ell$ where $g(y)$ is a scalar function of $y$. For LSR and LR learners, we have $g(y) = \frac{\langle\boldsymbol{w},\boldsymbol{x}\rangle-y}{\langle\boldsymbol{w},\boldsymbol{x}\rangle-\tilde{y}}$ and $g(y) = \frac{y(1+\exp(\tilde{y}\langle\boldsymbol{w},\boldsymbol{x}\rangle))}{\tilde{y}(1+\exp(y\langle\boldsymbol{w},\boldsymbol{x}\rangle))}$, respectively ($\tilde{y}$ is the original label for $\boldsymbol{x}$). In general, the new gradient of linear learners (*e.g.*, LSR, LR and SVM) in greedy LAST can be re-written as $g(y)\nabla_{\boldsymbol{w}}\ell$ where $g(y)$ varies for different cases. However, in the mini-batch case, modifying labels will become much more flexible than adjusting the learning rate, since $g(y)$ is generally sample-specific and learning rate is not.

We discuss how $g(y)$ determines the exponential teachability, since LAST can be viewed as modifying $g(y)$ to minimize the model parameter discrepancy. Here we consider the learner to have a loss function $f(\boldsymbol{w}) = \sum_{i=1}^{n}\ell_i(\boldsymbol{w})$ where $\ell_i(\boldsymbol{w}) = \ell(\boldsymbol{x}_{[i]}, y_{[i]}|\boldsymbol{w})$ and $n$ is the number of samples. We show in Theorem 1 that exponential teachability [43, 44] (*i.e.*, the teacher guides the learner to converge to $\boldsymbol{w}^*$ at an exponential speed) can be achieved by intelligently tuning $g(y)$ in LAST:

**Theorem 1** (Exponential teachability). *Assume the learner loss $\ell_i$ has the property of interpolation, $L_i$-Lipschitz, and convexity. $f$ is order-1 $\mu$ strongly convex. Then LAST can achieve ET with $g(y) = c_1\|\boldsymbol{w}^t - \boldsymbol{w}^*\|$, i.e., $\mathbb{E}\{\|\boldsymbol{w}^T - \boldsymbol{w}^*\|^2\} \leq (1 - c_1\eta_t\bar{\mu} + c_1^2\eta_t^2 L_{\max})^T \|\boldsymbol{w}^0 - \boldsymbol{w}^*\|^2$ where $L_{\max} = \max_i L_i$ and $\bar{\mu} = \sum_i\mu_i/n$. It implies that $\mathcal{O}((\log\frac{1}{c_0})^{-1}\log(\frac{1}{\epsilon}))$ samples are needed to achieve $\mathbb{E}\{\|\boldsymbol{w}^T - \boldsymbol{w}^*\|^2\} \leq \epsilon$. $c_0 = 1 - c_1\eta_t\bar{\mu} + c_1^2\eta_t^2 L_{\max}$ and $c_1$ is adjusted such that $0 < c_1\eta_t < \bar{\mu}/L_{\max}$.*

Theorem 1 shows that $g(y)$ plays a critical role, similar in spirit to the way that in line search, the step size is adaptively adjusted. We emphasize that a random teacher (*i.e.*, vanilla SGD) cannot achieve ET and yields an optimal rate of $\mathcal{O}(\frac{1}{\epsilon})$ [60] (in order to reach $\epsilon$-approximation of loss value). Additionally, we connect Theorem 1 to Armijo linear search [6, 80] by showing that ET can also be achieved by tuning $g(y)$ in a more fine-grained way under a different set of conditions:

**Theorem 2** (Exponential teachability without target parameters). *Assume that the learner loss $\ell_i$ has the property of interpolation, $L_i$-smoothness, convexity and $f$ has the property of $\mu$ strong-convexity of order 2. If we adjust $g(y)$ such that the following condition is always satisfied in each iteration: $\ell_{i_t}(\boldsymbol{w}^t - \eta_t g(y)\nabla\ell_{i_t}(\boldsymbol{w}^t)) \leq \ell_{i_t}(\boldsymbol{w}^t) - c_2\eta_t g(y)\|\nabla\ell_{i_t}(\boldsymbol{w}^t)\|^2$, then LAST can achieve ET for iterative learners with $c_2 = \frac{1}{2}$: $\mathbb{E}\{\|\boldsymbol{w}^T - \boldsymbol{w}^*\|^2\} \leq \max\left\{(1 - \frac{\bar{\mu}}{L_{\max}}), (1 - \bar{\mu}\eta'_{\max})\right\}^T \|\boldsymbol{w}^0 - \boldsymbol{w}^*\|^2$ which indicates that $\mathcal{O}(c_3\log(\frac{1}{\epsilon}))$ samples are needed to achieve $\mathbb{E}\{\|\boldsymbol{w}^T - \boldsymbol{w}^*\|^2\} \leq \epsilon$.*

**Remark 1** (Meaningfulness of omniscient teaching). *A natural question arises: what is the point of the target parameters $\boldsymbol{w}^*$ if we can achieve ET without it? First of all, both Theorem 1 and Theorem 2 merely consider the case where the synthesized label is a scaler, while labels are usually high-dimensional vectors in practice. When labels are vectors, LAST is much more flexible than tuning the learning rate. Second, the advantages of LAST is reflected in the easiness of setting $g(y)$. Because there exists a universal constant $c_1$ for $g(y) = c_1\|\boldsymbol{w}^t - \boldsymbol{w}^*\|$, $g(y)$ is much easier to set in Theorem 1. In contrast, $g(y)$ in Theorem 2 has to satisfy a dynamically changing inequality, which is highly nontrivial. Third, Theorem 2 also shows that Theorem 1 can still be substantially improved and the current lower bound on iterative teaching dimension is not necessarily tight.*

When the label $y$ is a scalar, LAST can also be viewed as a method that intelligently controls the step size in order to improve convergence to a desired model $\boldsymbol{w}^*$. Theorem 1 is reasonably comparable to the theoretical results in [43, 44], but it does not require to modify $\boldsymbol{x}$ to achieve ET. In contrast to deterministic ET in [43], LAST achieves probabilistic ET due to the randomness in data sampling. When $y$ becomes a vector, then LAST essentially controls a multi-dimensional learning rate that can modify the gradient in a more fine-grained fashion. As an example, for LSR learners with a vector $\boldsymbol{y}\in\mathbb{R}^K$, we have the new gradient matrix of size $d \times K$ (after label synthesis) as $\nabla'_{\boldsymbol{w}}\ell = g(\boldsymbol{y})\cdot\nabla_{\boldsymbol{w}}\ell$ where $g(\boldsymbol{y}) = \frac{1}{d}(\boldsymbol{y} - \langle\boldsymbol{w},\boldsymbol{x}\rangle)\cdot(1\oslash(\tilde{\boldsymbol{y}} - \langle\boldsymbol{w},\boldsymbol{x}\rangle))$ ($\oslash$ denotes the Hadamard division). More interestingly, Theorem 2 also validates the feasibility of black-box LAST teaching, since the convergence can be improved even without target parameters. We show in Theorem 3 that super-exponential teachability can be achieved under certain conditions and a carefully chosen $g(\boldsymbol{y})$:

**Theorem 3** (Super-exponential teachability). *Consider a learner loss $f(\boldsymbol{w}) = \sum_{i=1}^{n}\ell_i(\boldsymbol{w})$ and an optimal weight matrix $\boldsymbol{w}^*$ such that $\nabla_{\boldsymbol{w}}f(\boldsymbol{w}^*) = 0$. Assume that $\nabla_{\boldsymbol{w}}f(\boldsymbol{w})$ is $L$-smooth and continuously differentiable within a sufficiently small $\delta$-neighborhood $\|\boldsymbol{w} - \boldsymbol{w}^*\| \leq \delta$. In LAST, the gradient update is $\boldsymbol{w}^{t+1} = \boldsymbol{w}^t - \eta_t g(\boldsymbol{y})\nabla_{\boldsymbol{w}}f(\boldsymbol{w})$. If we have that $\boldsymbol{w}^0$ is in the $\delta$-neighborhood, $|\lambda_{\min}(\nabla_{\boldsymbol{w}}^2 f(\boldsymbol{w}))| \geq \mu$ and $L\mu\delta < 2$, then LAST can achieve super-ET with $g(\boldsymbol{y}) = \eta_t^{-1}(\nabla_{\boldsymbol{w}}^2 f(\alpha\boldsymbol{w}^* + (1-\alpha)\boldsymbol{w}))^{-1}$ where $\alpha\in[0,1]$. When $\alpha = 0$, the result reduces to Newton's method.*

Theorem 3 can be viewed as a generalization of Newton's method. In contrast to Newton's method, LAST takes advantages of the knowledge of the optimal learner $\boldsymbol{w}^*$ and reduces the requirement of the Hessian information. If we simply set $\alpha = 1$, then LAST only needs a single static $\nabla^2_{\boldsymbol{w}} f(\boldsymbol{w}^*)$ to achieve super-ET. In contrast, Newton's method requires $\nabla^2_{\boldsymbol{w}} f(\boldsymbol{w})$ in every iteration (*i.e.*, Hessian information of a local neighborhood). In the light of [64], we can easily extend Theorem 3 to work with sub-sampled Hessian $\frac{1}{|\mathcal{S}|} \sum_{i \in |\mathcal{S}|} \nabla^2_{\boldsymbol{w}} \ell_i(\boldsymbol{w}^*)$ where $\mathcal{S}$ is a subset of indices.

### 3.5 The Best of Both Worlds? Combining Sample Selection to LAST

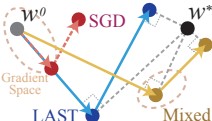

Figure 5: Comparison.

It is natural to further consider combining sample selection with LAST. A straightforward approach is to first perform the teaching sample selection in IMT and then run LAST to synthesize the labels for the selected samples. This can be viewed as applying alternating optimization to approximately solve $\min_{\boldsymbol{x},\boldsymbol{y}} G(\boldsymbol{x}, \boldsymbol{y}|\boldsymbol{w}^t)$. Specifically, mixed teaching first solves $\boldsymbol{x}$ with $\boldsymbol{y}$ fixed as the original ground truth label, and then solves $\boldsymbol{y}$ with $\boldsymbol{x}$ fixed. This procedure is iterated until sufficiently small error $\|\boldsymbol{w}^* - \boldsymbol{w}^t\|$ is attained. Due to stronger flexibility, this mixed greedy teaching strategy yields a more powerful policy than LAST. Based on [43], it is easy to verify that mixed teaching can achieve ET. Fig. 5 gives an illustrative comparison of SGD, LAST and mixed teaching. Unlike LAST that tends to follow the largest gradient direction (same as SGD), mixed teaching finds the closest point to $\boldsymbol{w}^*$ in the gradient space. Mixed teaching combines label synthesis in LAST with example selection in IMT, and is able to achieve faster teaching empirically.

## 4  A Distant Look at Black-box Label Synthesis Teaching

The previous section considers omniscient teaching where everything about the learner is accessible. However, the optimal learner parameters ($\boldsymbol{w}^*$) are often unavailable in practice. This motivates us to study black-box teaching [18, 19, 44, 88] where the teacher knows much less information about the learner. This problem is extremely challenging, but for completeness, we briefly discuss some potential strategies for black-box LAST (BLAST) to tackle this difficult problem and also provide some preliminary experiments in Appendix F.8. Because a black-box teacher has no access to $\boldsymbol{w}^*$, we need to use a surrogate objective to efficiently guide the learner to $\boldsymbol{w}^* = \arg\min_{\boldsymbol{w}} \mathbb{E}_{(\boldsymbol{x},\boldsymbol{y})}\{\ell(\boldsymbol{x}, \tilde{\boldsymbol{y}}|\boldsymbol{w})\}$. Such a surrogate objective can either be a differentiable criterion on a validation set (*e.g.*, some learner loss function) or a non-differentiable performance metric on a validation set (*e.g.*, validation accuracy). For the former, we can unroll the teacher into the optimization of the learner. For the latter, we can define the performance metric as the reward signal and formulate black-box teaching as a reinforcement learning problem. Then we utilize the policy gradient method to solve it.

**Unrolling**. We start with a fully differentiable variant for BLAST. Because $\boldsymbol{w}^*$ is no longer available in the black-box scenario, the objective $\|\boldsymbol{w}^v - \boldsymbol{w}^*\|$ becomes infeasible. In order to conduct the black-box teaching, we propose to directly use the learner loss $\ell(\boldsymbol{x}, \tilde{\boldsymbol{y}}|\boldsymbol{w})$ as the surrogate. After removing the dependence on $\boldsymbol{w}^*$ from Eq. (5), we obtain the objective function of BLAST:

$$\min_{\boldsymbol{\theta}} \mathbb{E}_{\{\boldsymbol{x},\tilde{\boldsymbol{y}}\} \sim \mathcal{D}_a}\{\ell(\boldsymbol{x}, \tilde{\boldsymbol{y}}|\boldsymbol{w}^v)\} \quad \text{s.t. } \boldsymbol{w}^v(\boldsymbol{\theta}) = \arg\min_{\boldsymbol{w}} \mathbb{E}_{\{\boldsymbol{x},\tilde{\boldsymbol{y}}\} \sim \mathcal{D}_r}\{\ell(\boldsymbol{x}, \pi_{\boldsymbol{\theta}}(\boldsymbol{x}, \tilde{\boldsymbol{y}}, \boldsymbol{w}, \boldsymbol{w}^*)|\boldsymbol{w})\} \quad (6)$$

where $\tilde{\boldsymbol{y}}$ denotes the ground truth label of sample $\boldsymbol{x}$ and the output of $\pi_{\boldsymbol{\theta}}$ is the synthesized label $\boldsymbol{y}$. $\mathcal{D}_a$ denotes the validation set, and $\mathcal{D}_r$ denotes the training set. Sometimes we can use the same set as $\mathcal{D}_a$ and $\mathcal{D}_r$. Here we use stochastic gradient descent to solve the inner minimization. Typically we can unroll $v$-step gradient updates and train the teacher in an end-to-end fashion. To reduce the difficulty of learning the teacher in practice, it can be beneficial to parameterize the synthesized label to be $\alpha\tilde{\boldsymbol{y}} + (1-\alpha)\boldsymbol{y}'$ where $\alpha \in (0, 1)$ and $\boldsymbol{y}' = \pi_{\boldsymbol{\theta}}(\boldsymbol{x}, \tilde{\boldsymbol{y}}, \boldsymbol{w}, \boldsymbol{w}^*)$. This is essentially to learn a residue between the synthesized label and the ground truth label. The intuition behind is to constrain the space of synthesized labels such that the black-box teaching can be easier.

**Policy gradient**. We also discuss a more general variant for BLAST. We first frame black-box teaching as a reinforcement learning task. Since we have no access to the optimal model $\boldsymbol{w}^*$, we use a hold-out set to evaluate the performance of the learner (Fig. 6), which also serves as the surrogate learning objective. Typically the performance on the hold-out set is non-differentiable *w.r.t.* $\boldsymbol{\theta}$, so we leverage the policy gradient method [86] to learn the teaching policy $\boldsymbol{\theta}$. Specifically, we use the following objective function: $\max_{\boldsymbol{\theta}} J(\boldsymbol{\theta}) := \mathbb{E}_{(s_t,a_t) \sim \pi_{\boldsymbol{\theta}}(a|s)}\{\sum_{t=1}^{T} r_t(s_t, a_t)\}$ where $T$ denotes the termination iteration, $(s_t, a_t)$ is the state-action pair at the $t$-the iteration and $r_t(s_t, a_t)$ is the reward. Since we use a terminal reward $R = r_T(s_t, a_t)$ defined on a hold-out set, the objective becomes $J(\boldsymbol{\theta}) = \mathbb{E}_{\pi_{\boldsymbol{\theta}}}\{R\}$. Therefore, we update the teacher parameters $\boldsymbol{\theta}$ with $\boldsymbol{\theta} \leftarrow \boldsymbol{\theta} + \eta \cdot \nabla_{\boldsymbol{\theta}} J(\boldsymbol{\theta})$ where $\nabla_{\boldsymbol{\theta}} J(\boldsymbol{\theta}) = \sum_t \nabla_{\boldsymbol{\theta}} \log \pi_{\boldsymbol{\theta}}(a_t|s_t)R$. We take classification as an illustrative example here.

Specifically, there are many ways to define the terminal reward, for example, *(i)* the accuracy after $T$ iterations on the hold-out set, and *(ii)* the number of iterations to achieve a preset accuracy $\zeta$ on the hold-out set ($T$ is the maximal iteration number). We can parameterize the teacher as any approximation method, such as linear classifier or neural network. The learner optimizes its weights by cross-entropy loss and SGD. The teaching policy aims to generate suitable labels for input samples so that the learner can converge efficiently. Fig. 6 illustrates the reinforcement learning formulation. Since there are many choices to design

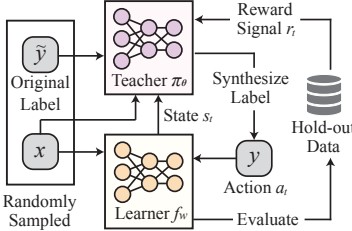

Figure 6: RL formulation for BLAST.

state features, action space and teaching policy, we aim to demonstrate the feasibility of applying reinforcement learning to perform black-box teaching instead of enumerating all the design choices.

- **State**. State features are the input for the teacher model and are crucial for learning a good teaching policy. As an example for linear learners, we can construct the state features $h(s_t)$ with the current learner $\boldsymbol{w}$, the optimal learner $\boldsymbol{w}^*$, the input sample $\boldsymbol{x}$ and the original ground truth label $\tilde{\boldsymbol{y}}$. For nonlinear learners, it remains an open problem to design informative state features [7, 18, 37].

- **Action**. For small number of classes, we can uniformly discretize the label space (*i.e.*, simplex) into a set of vectors. For example, the binary label space can be discretized into $[0, 1]$, $[0.25, 0.75]$, $[0.5, 0.5]$, $[0.75, 0.25]$ and $[1, 0]$. For large number of classes, we need to reduce search difficulty in the label space. For example, inspired by label smoothing (LS) and knowledge distillation (KD), we can design the synthesized label to be $\boldsymbol{y} = \mu \tilde{\boldsymbol{y}} + (1 - \mu) \boldsymbol{p}$ where $\tilde{\boldsymbol{y}}$ is the one-hot ground truth label. $\boldsymbol{p}$ is an all-$\frac{1}{K}$ vector in LS and is a soft label vector from a pretrained model in KD ($K$ is the number of classes). We discretize the action space $\mu \in [0.5, 1]$ and use the reward to guide the search of state-dependent $\mu$. The teaching policy produces $\mu_t = \pi_{\boldsymbol{\theta}}(h(s_t))$ where $\pi_{\boldsymbol{\theta}}$ can be a MLP.

## 5 Beyond LAST: Intriguing Insights into Iterative Teaching

### 5.1 Additional Discussions

**Practicality of iterative teaching**. One of the most significant differences between batch machine teaching (BMT) and IMT is whether learner's optimization algorithm is taken into consideration or not. More generally, IMT views the learner as a specific procedure or algorithm, while BMT views the learner as an objective function. When the objective function becomes non-convex and has multiple equivalent global minima, then it could be infeasible for BMT to construct a minimal teaching set for guiding the learner to a particular target model $\boldsymbol{w}^*$. In contrast, IMT can still guide the learner to a specific model by considering the parameter update in every iteration. Overall, IMT makes a paradigm shift from focusing on the learner's model to focusing on the learner's algorithm.

**Iterative teaching as constrained communication**. We provide an alternative perspective to look at iterative teaching. If we view the teacher as a sender and the learner as a receiver, then the task of iterative teaching is to transmit some information (*i.e.*, the target model $\boldsymbol{w}^*$) from the sender to the receiver. Such information can only be transmitted under a constrained channel. For example, vanilla IMT [43] uses a communication channel that can only transmit teaching examples and LAST uses a channel that only transmits labels. The teacher encodes the target model into teaching examples or labels, and the learner uses a decoder (*i.e.*, its learning algorithm) to recover the target model. The constraints on the communication channel determine the difficulty of iterative teaching.

**What makes iterative teaching interesting?** First of all, iterative teaching provides a theoretically grounded framework to study how different sequences of training samples (*i.e.*, different data sampling schemes) affects the convergence of a learner. More broadly, iterative teaching could help us better understand to what extent the inductive bias can be affected by different sequences of training samples. This has immediate connections to curriculum learning [7], self-paced learning [37] and importance sampling [99]. Second, iterative teaching can inspire new heuristics to improve convergence and generalization for complex nonlinear learners (*e.g.*, neural networks) in practice. For example, both vanilla IMT and LAST imply that training a learner with easy samples first and hard samples later can improve convergence. Such an implication can be used to design practical training heuristics. Moreover, black-box iterative teaching is studied to bridge the gap between omniscient iterative teaching and practical machine learning. Third, iterative teaching yields a unified framework where the definition of teaching medium is very flexible. We have studied sample selection [43, 44, 91], label synthesis (this paper) and the combination of both ([43] and this paper). Iterative teaching is not limited to training data and the effectiveness of other teaching spaces remains to be explored.

**Open problems**. Iterative teaching is still in its infancy and there are numerous open problems in its theory, algorithm and practice. From the theory side, we can only obtain a lower bound for the iterative teaching dimension for now. The tightness of such lower bounds remains unknown. Moreover, a upper bound for the iterative teaching dimension and how to loosen assumptions on learners are also important open problems. From the algorithm side, better algorithms to perform omniscient, gray-box or black-box teaching are yet to be designed. In practice, the scalability and efficiency of iterative teaching is also a major bottleneck for large-scale applications [91].

## 5.2 A Generalized Iterative Teaching Framework

We present a generalized iterative teaching (GIT) framework that considers a broader class of iterative learners (not limited to first-order gradient learners). At the $t$-th iteration, GIT is formulated as

$$\boxed{\text{Learner:}} \; \boldsymbol{w}^{t+1} = \underbrace{\mathcal{Q}\big(\{\mathcal{M}_i^{[t]}\}_{i=1}^m, \boldsymbol{w}^t, \ell\big)}_{\text{Iterative update function}} \qquad \boxed{\text{Teacher:}} \; \underset{\{\{\mathcal{M}_i^{[1]}\}_{i=1}^m, \cdots, \{\mathcal{M}_i^{[v]}\}_{i=1}^m\}}{\arg\min} \; d\big(\boldsymbol{w}^{t+v}, \boldsymbol{w}^*\big) \quad (7)$$

where $\mathcal{M}$ denotes the teaching material (*e.g.*, $\mathcal{M}_i^{[j]} = \{\boldsymbol{x}_i^j, \boldsymbol{y}_i^j\}$ is the $i$-th training sample in the mini-batch of the $j$-th iteration), $\mathcal{Q}$ is a general function to denote learner's iterative update rule (*e.g.*, SGD learner uses $\mathcal{Q} = \boldsymbol{w}^t - \eta_t \ell'(\boldsymbol{x}^t, \boldsymbol{y}^t | \boldsymbol{w}^t)$), $\ell$ is the learner's objective function, $\boldsymbol{w}^t$ is the learner's model parameters at the $t$-th iteration and $\boldsymbol{w}^*$ is the target parameters. The teacher interacts with the learner every $v$ iterations. Then we denote this generalized teaching problem as GIT$(m, v, \mathcal{M}, \mathcal{Q})$. For standard iterative teaching, we use training data as $\mathcal{M}$ and the SGD update as $\mathcal{Q}$ and abbreviate the problem as GIT$(m, v)$. For the ultimate IMT problem, we aim to solve GIT$(m, T)$ where $T$ is the termination iteration. Most prior studies [43, 44, 91] focus on solving GIT$(1, 1)$, while our paper addresses GIT$(1, v)$ where $v \geq 1$. For strongly convex learners, BMT is equivalent to finding a minimal $m^*$ that solves GIT$(m^*, \infty)$ under the constraint of $\mathcal{M}_i^{[1]} = \mathcal{M}_i^{[j]}, \forall j$, and $m^*$ is also the teaching dimension in BMT. Alternatively, BMT also solves GIT$(m^*, \infty, \{\boldsymbol{x}, \boldsymbol{y}\}, \mathcal{Q}(\{\mathcal{M}_i^{[1]}\}_{i=1}^{m^*}, \boldsymbol{w}, \ell))$ where the learner always uses the mini-batch in the first iteration to update itself (and ignores mini-batches in the other iteration). The first mini-batch is the same as the minimal teaching set in BMT.

## 6 Empirical Evaluation

This section comprehensively evaluates LAST in the omniscient teaching scenario. Experimental details and more results (including BLAST) are given in Appendix G and Appendix F, respectively.

### 6.1 Omniscient Greedy Teaching

**Least square regression learner**. We evaluate LAST on linear regression where the training data is generated by $\tilde{\boldsymbol{y}} = \langle \boldsymbol{w}^*, \boldsymbol{x} \rangle + \epsilon$ ($\epsilon$ follows the normal distribution). We compute the training objective value and the Euclidean distance between $\boldsymbol{w}^t$ and $\boldsymbol{w}^*$ in Fig. 7 to evaluate empirical convergence. For LAST, we test "no constraint" (denoted as "NC") and "magnitude constraint". The

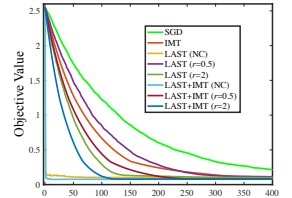 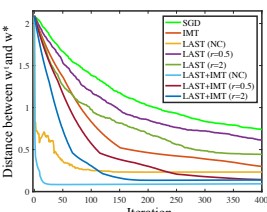

Figure 7: Convergence curves for linear regression. LAST $(r=0.5)$ denotes LAST with magnitude constraint of $r = 0.5$.

results show that LAST generally converges better with a more flexible teacher, and outperforms IMT with magnitude coefficient $r = 2$. This suggests that a suitable label may sometimes be more crucial than a suitable input example. We also observe that mixed teaching (LAST+IMT) converges significantly faster than IMT or LAST alone, implying that alternating optimization works well.

**Logistic regression learner**. Next we evaluate LAST on linear classification for logistic regression learners. We teach the logistic regression learner to perform binary classification on both synthetic data and real image data. For synthetic data, we use 2D half moon data and Gaussian cluster data (in Appendix F). For real image data, we use 3/5 digits in MNIST. From Fig. 8, we observe that mixed teaching converges efficiently and consistently achieves the best performance.

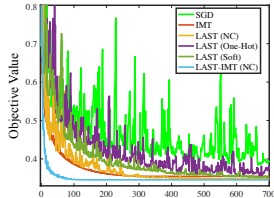 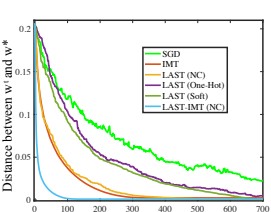

Figure 8: Binary classification for logistic regression. One-Hot: one-hot constraint. Soft: soft constraints.

All variants of LAST significantly outperform SGD. Without constraints, LAST is comparable to IMT in terms of convergence but does not require any costly example selection.

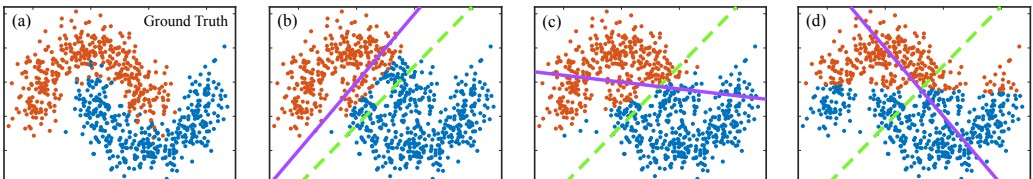

Figure 9: Synthesized labels on half-moon data with 1-hot constraint. (a) ground truth labels. (b,c,d) synthesized labels by LAST. The purple line is the current classifier and the green dotted line is the optimal classifier.

Fig. 9 shows how synthesized labels change depending on the learner's parameters. We observe that LAST synthesizes labels adaptively based on the current learner. Our results connect well to the VC theory that the convergence rate depends on how well a classifier can separate the data [77]. The synthesized labels are always linearly separable, which effectively improves convergence.

**Multi-layer perceptron learner**. We apply LAST to neural networks in order to empirically show its effectiveness for nonlinear learners. Specifically, we use LAST ($\beta = 1$) to teach a 2-layer MLP with ReLU nonlinearity (the output layer is essentially a logistic regression classifier). We perform binary classification of digit 3 and 5 on MNIST. Results in Fig. 10 show that all the variants of LAST achieve significantly better conver-

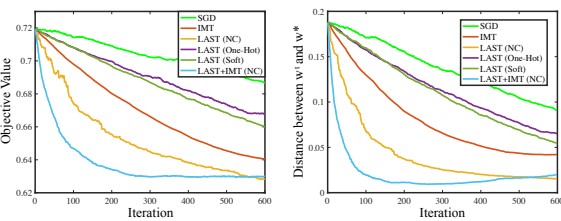

Figure 10: Binary classification for MLP learners.

gence than SGD. With a teacher that generates labels without constraints, both LAST and mixed teaching achieve far better convergence than IMT. LAST is also much more efficient than IMT.

Fig. 11 presents some teaching examples for MLP learners in 3/5 MNIST digit classification, where the one-hot labels synthesized by LAST differ from the ground truth. The synthesized labels depend on the learner status and may vary across different iterations. We can observe that most of the examples whose synthesized labels are different from the ground truth ones are semantically ambiguous to humans, suggesting that labels of hard samples near the decision boundary are very crucial for convergence. Moreover, such a observation is consistent for linear learners (see Appendix F.4).



Figure 11: Ground truth vs. LAST.

## 6.2 Omniscient Parameterized Teaching

We learn a parameterized teaching policy for logistic regression classifier learners using both unrolling and policy gradient. We parameterize the teacher model with a MLP. Experimental details are given in Appendix G. We evaluate the learned policies on binary classification of digit 3/5 in MNIST. Fig. 12 shows that both unrolling and policy gradient can learn an effective teaching policy that achieves better convergence than SGD. Moreover, we evaluate two

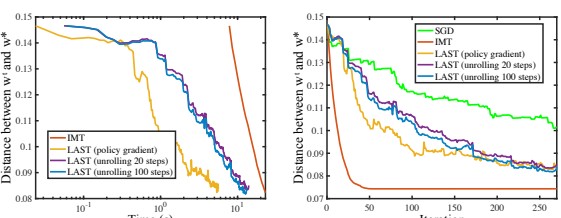

Figure 12: Omniscient parameterized teaching on Binary classification for logistic regression learners. Left: time-wise convergence. Right: iteration-wise convergence.

unrolling settings (20 steps and 100 steps) and unrolling more steps show marginal benefits on convergence. On time-wise convergence, we can observe that all variants of LAST performs significantly faster than IMT, because no costly example selection is needed. On iteration-wise convergence, we can see that IMT still achieves the fastest convergence due to its strong flexibility from example selection. LAST with either unrolling or policy gradient converges much faster than SGD, and policy gradient yields slightly better convergence than unrolling in the experiments.

## 7  Concluding Remarks

In this paper, we present a novel iterative teaching framework via label synthesis. We consider both the omniscient scenario where the teacher has access to the optimal learner parameters and the black-box scenario where the teacher has no access to the optimal learner parameters. We propose greedy teaching and parameterized teaching, and provide theoretical insights on why they can guide the learner to converge quickly. Extensive experiments demonstrate the effectiveness of our methods.

## Acknowledgments

WL is supported by a Cambridge-Tübingen Fellowship, an NVIDIA GPU grant, DeepMind and the Leverhulme Trust via CFI. AW acknowledges support from a Turing AI Fellowship under grant EP/V025379/1, The Alan Turing Institute, and the Leverhulme Trust via CFI. LP is supported by CIFAR AI Chair Program and acknowledges support from Samsung Electronics Co., Ldt. This work was supported by the German Federal Ministry of Education and Research (BMBF): Tübingen AI Center, FKZ: 01IS18039B, and by the Machine Learning Cluster of Excellence, EXC number 2064/1 – Project number 390727645.

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
