# Appendix

**Table of Contents**

# A  Proof of Proposition 1

*Proof.* This proposition is quite intuitive. For the first iteration, the SGD uses the following update:

$$\boldsymbol{w}_s^1 = \boldsymbol{w}_0 - \eta \frac{\partial \ell(\boldsymbol{x}_{i_1}, y_{i_1} | \boldsymbol{w})}{\partial \boldsymbol{w}} \tag{8}$$

which gives the following distance to $\boldsymbol{w}^*$:

$$
\begin{aligned}
\left\| \boldsymbol{w}_s^1 - \boldsymbol{w}^* \right\|^2 &= \left\| \boldsymbol{w}_0 - \eta \frac{\partial \ell(\boldsymbol{x}_{i_1}, \tilde{y}_{i_1} | \boldsymbol{w})}{\partial \boldsymbol{w}} - \boldsymbol{w}^* \right\|^2 \\
&= \left\| \boldsymbol{w}^0 - \boldsymbol{w}^* \right\|^2 + T_{12}(\boldsymbol{x}_{i_1}, \tilde{y}_{i_1} | \boldsymbol{w}^0)
\end{aligned}
\tag{9}
$$

where $i_t, \forall t$ is uniformly sampled index and we define $T_{12}(\boldsymbol{x}_{i_1}, \tilde{y}_{i_1} | \boldsymbol{w}^0) = \eta^2 T_1(\boldsymbol{x}_{i_1}, \tilde{y}_{i_1} | \boldsymbol{w}^0) - 2\eta T_2(\boldsymbol{x}_{i_1}, \tilde{y}_{i_1} | \boldsymbol{w}^0)$. With the same randomly sampled teaching data point $\{\boldsymbol{x}_{i_1}, \tilde{y}_{i_1}\}$, the greedy LAST teacher yields the following distance to $\boldsymbol{w}^*$:

$$
\begin{aligned}
\left\| \boldsymbol{w}_l^1 - \boldsymbol{w}^* \right\|^2 &= \left\| \boldsymbol{w}_0 - \eta \frac{\partial \ell(\boldsymbol{x}_{i_1}, y_{i_1} | \boldsymbol{w})}{\partial \boldsymbol{w}} - \boldsymbol{w}^* \right\|^2 \\
&= \left\| \boldsymbol{w}^0 - \boldsymbol{w}^* \right\|^2 + \min_{y'_{i_1}} T_{12}(\boldsymbol{x}, y'_{i_1} | \boldsymbol{w}^0).
\end{aligned}
\tag{10}
$$

where $y_{i_1} = \arg\min_{y'} T_{12}(\boldsymbol{x}, y'_{i_1} | \boldsymbol{w}^0)$. Therefore, we have that $\left\| \boldsymbol{w}_l^1 - \boldsymbol{w}^* \right\| \leq \left\| \boldsymbol{w}_s^1 - \boldsymbol{w}^* \right\|$ for any $i_1$, which leads to $\mathbb{E}_{i_1}\{\left\| \boldsymbol{w}_l^1 - \boldsymbol{w}^* \right\|\} \leq \mathbb{E}_{i_1}\{\left\| \boldsymbol{w}_s^1 - \boldsymbol{w}^* \right\|\}$. We consider the second iteration for SGD:

$$
\begin{aligned}
\mathbb{E}_{i_1, i_2}\{\left\| \boldsymbol{w}_s^2 - \boldsymbol{w}^* \right\|^2\} &= \mathbb{E}_{i_1, i_2}\{G(\boldsymbol{x}_{i_2}, y'_{i_2} | \boldsymbol{w}_s^1)\} \\
&\geq \mathbb{E}_{i_1, i_2}\{G(\boldsymbol{x}_{i_2}, y'_{i_2} | \boldsymbol{w}_l^1)\} \\
&= \mathbb{E}_{i_1, i_2}\{\left\| \boldsymbol{w}_l^2 - \boldsymbol{w}^* \right\|^2\}
\end{aligned}
\tag{11}
$$

which yields $\mathbb{E}_{i_1, i_2}\{\left\| \boldsymbol{w}_l^2 - \boldsymbol{w}^* \right\|^2\} \leq \mathbb{E}_{i_1, i_2}\{\left\| \boldsymbol{w}_s^2 - \boldsymbol{w}^* \right\|^2\}$. For the third iteration, we have for SGD that

$$
\begin{aligned}
\mathbb{E}_{i_1, i_2, i_3}\{\left\| \boldsymbol{w}_s^3 - \boldsymbol{w}^* \right\|^2\} &= \mathbb{E}_{i_1, i_2, i_3}\{G(\boldsymbol{x}_{i_2}, y'_{i_2} | \boldsymbol{w}_s^2)\} \\
&\geq \mathbb{E}_{i_1, i_2, i_3}\{G(\boldsymbol{x}_{i_2}, y'_{i_2} | \boldsymbol{w}_l^2)\} \\
&= \mathbb{E}_{i_1, i_2, i_3}\{\left\| \boldsymbol{w}_l^3 - \boldsymbol{w}^* \right\|^2\}.
\end{aligned}
\tag{12}
$$

Eventually, we will have that $\mathbb{E}\{\left\| \boldsymbol{w}_l^T - \boldsymbol{w}^* \right\|^2\} \leq \mathbb{E}\{\left\| \boldsymbol{w}_s^T - \boldsymbol{w}^* \right\|^2\}$ at the $T$-th iteration, which means that greedy LAST converges no slower than SGD. □

# B Proof of Theorem 1

From the $(t+1)$-th gradient update with the greedy LAST teacher (for $\boldsymbol{x}_i$, $y_i$ is the synthesized label and $\tilde{y}_i$ is the ground truth label), we have that

$$
\begin{aligned}
\left\|\boldsymbol{w}^{t+1} - \boldsymbol{w}^*\right\|^2 &= \left\|\boldsymbol{w}^t - \eta_t \nabla_{\boldsymbol{w}^t}\ell(\boldsymbol{x}_{i_t}, y_{i_t}|\boldsymbol{w}^t) - \boldsymbol{w}^*\right\|^2 \\
&= \left\|\boldsymbol{w}^t - \eta_t g(y_{i_t})\nabla_{\boldsymbol{w}^t}\ell(\boldsymbol{x}_{i_t}, \tilde{y}_{i_t}|\boldsymbol{w}^t) - \boldsymbol{w}^*\right\|^2 \\
&= \left\|\boldsymbol{w}^t - \boldsymbol{w}^*\right\|^2 - 2\eta_t g(y_{i_t})\langle\nabla_{\boldsymbol{w}^t}\ell(\boldsymbol{x}_{i_t}, \tilde{y}_{i_t}|\boldsymbol{w}^t), \boldsymbol{w}^t - \boldsymbol{w}^*\rangle \\
&\quad + \eta_t^2(g(y_{i_t}))^2\left\|\nabla_{\boldsymbol{w}^t}\ell(\boldsymbol{x}_{i_t}, \tilde{y}_{i_t}|\boldsymbol{w}^t)\right\|^2
\end{aligned}
\tag{13}
$$

where $i_t$ denotes a randomly sampled index from the pool in the $t$-th iteration. It can be simplified as (by denoting $\nabla_{\boldsymbol{w}^t}\ell(\boldsymbol{x}_{i_t}, \tilde{y}_{i_t}|\boldsymbol{w}^t)$ as $\nabla\ell_{i_t}(\boldsymbol{w}^t)$):

$$
\left\|\boldsymbol{w}^{t+1} - \boldsymbol{w}^*\right\|^2 = \left\|\boldsymbol{w}^t - \boldsymbol{w}^*\right\|^2 - 2\eta_t g(y_{i_t})\langle\nabla\ell_{i_t}(\boldsymbol{w}^t), \boldsymbol{w}^t - \boldsymbol{w}^*\rangle + \eta_t^2(g(y_{i_t}))^2\left\|\nabla\ell_{i_t}(\boldsymbol{w}^t)\right\|^2.
\tag{14}
$$

Because we know that the synthesized label $y_{i_t}$ is the solution to the following minimization:

$$
y_{i_t} = \arg\min_{y'_{i_t}}\left\{\eta_t^2(g(y'_{i_t}))^2\left\|\nabla_{\boldsymbol{w}^t}\ell(\boldsymbol{x}_{i_t}, \tilde{y}_{i_t}|\boldsymbol{w}^t)\right\|^2 - 2\eta_t g(y'_{i_t})\langle\nabla_{\boldsymbol{w}^t}\ell(\boldsymbol{x}_{i_t}, \tilde{y}_{i_t}|\boldsymbol{w}^t), \boldsymbol{w}^t - \boldsymbol{w}^*\rangle\right\},
\tag{15}
$$

then we plug a new $y''_{i_t}$ which satisfies $g(y''_{i_t}) = c_1\|\boldsymbol{w}^t - \boldsymbol{w}^*\|$ to Eq. (14) and have the following inequality:

$$
\begin{aligned}
\left\|\boldsymbol{w}^{t+1} - \boldsymbol{w}^*\right\|^2 &\leq \left\|\boldsymbol{w}^t - \boldsymbol{w}^*\right\|^2 - 2\eta_t g(y''_{i_t})\langle\nabla\ell_{i_t}(\boldsymbol{w}^t), \boldsymbol{w}^t - \boldsymbol{w}^*\rangle + \eta_t^2(g(y''_{i_t}))^2\left\|\nabla\ell_{i_t}(\boldsymbol{w}^t)\right\|^2 \\
&= \left\|\boldsymbol{w}^t - \boldsymbol{w}^*\right\|^2 - 2\eta_t c_1\left\|\boldsymbol{w}^t - \boldsymbol{w}^*\right\|\langle\nabla\ell_{i_t}(\boldsymbol{w}^t), \boldsymbol{w}^t - \boldsymbol{w}^*\rangle \\
&\quad + \eta_t^2 c_1^2\left\|\nabla\ell_{i_t}(\boldsymbol{w}^t)\right\|^2\left\|\boldsymbol{w}^t - \boldsymbol{w}^*\right\|^2
\end{aligned}
\tag{16}
$$

Using the convexity of $f(\cdot)$ and the order-1 strong convexity [40] of $\ell_{i_t}(\cdot)$, we have that (let $\mu_{i_t} = 0$ if $\ell_{i_t}$ is not order-1 strongly convex):

$$
-\langle\nabla\ell_{i_t}(\boldsymbol{w}^t), \boldsymbol{w}^t - \boldsymbol{w}^*\rangle \leq \ell_{i_t}(\boldsymbol{w}^*) - \ell_{i_t}(\boldsymbol{w}^t) - \frac{\mu_{i_t}}{2}\left\|\boldsymbol{w}^t - \boldsymbol{w}^*\right\|
\tag{17}
$$

which leads to

$$
\begin{aligned}
\left\|\boldsymbol{w}^{t+1} - \boldsymbol{w}^*\right\|^2 &\leq \left\|\boldsymbol{w}^t - \boldsymbol{w}^*\right\|^2 + 2\eta_t c_1\left\|\boldsymbol{w}^t - \boldsymbol{w}^*\right\|\left(\ell_{i_t}(\boldsymbol{w}^*) - \ell_{i_t}(\boldsymbol{w}^t) - \frac{\mu_{i_t}}{2}\left\|\boldsymbol{w}^t - \boldsymbol{w}^*\right\|\right) \\
&\quad + c_1^2\left\|\nabla\ell_{i_t}(\boldsymbol{w}^t)\right\|^2\left\|\boldsymbol{w}^t - \boldsymbol{w}^*\right\|^2 \\
&= \left\|\boldsymbol{w}^t - \boldsymbol{w}^*\right\|^2 + 2\eta_t c_1\left\|\boldsymbol{w}^t - \boldsymbol{w}^*\right\|(\ell_{i_t}(\boldsymbol{w}^*) - \ell_{i_t}(\boldsymbol{w}^t)) - \eta_t c_1\mu_{i_t}\left\|\boldsymbol{w}^t - \boldsymbol{w}^*\right\|^2 \\
&\quad + c_1^2\left\|\nabla\ell_{i_t}(\boldsymbol{w}^t)\right\|^2\left\|\boldsymbol{w}^t - \boldsymbol{w}^*\right\|^2.
\end{aligned}
$$

Using the condition that $\ell_i$ is $L_i$-Lipschitz continuous and denoting $L_{\max} = \max_i L_i$, we have that

$$
\begin{aligned}
\left\|\boldsymbol{w}^{t+1} - \boldsymbol{w}^*\right\|^2 &\leq \left\|\boldsymbol{w}^t - \boldsymbol{w}^*\right\|^2 + 2\eta_t c_1\left\|\boldsymbol{w}^t - \boldsymbol{w}^*\right\|(\ell_{i_t}(\boldsymbol{w}^*) \\
&\quad - \ell_{i_t}(\boldsymbol{w}^t)) - \eta_t c_1\mu_{i_t}\left\|\boldsymbol{w}^t - \boldsymbol{w}^*\right\|^2 + \eta_t^2 c_1^2 L_{\max}^2\left\|\boldsymbol{w}^t - \boldsymbol{w}^*\right\|^2.
\end{aligned}
$$

The interpolation condition [80] implies that $\boldsymbol{w}^*$ is the minimum for all functions $\ell_i$, which is equivalent to $\ell_i(\boldsymbol{w}^*) \leq \ell_i(\boldsymbol{w}^t)$ for all $i$. Therefore, we have that $(\ell_{i_t}(\boldsymbol{w}^*) - \ell_{i_t}(\boldsymbol{w}^t)) \leq 0$. Then we end up with

$$
\begin{aligned}
\left\|\boldsymbol{w}^{t+1} - \boldsymbol{w}^*\right\|^2 &\leq \left\|\boldsymbol{w}^t - \boldsymbol{w}^*\right\|^2 - \eta_t c_1\mu_{i_t}\left\|\boldsymbol{w}^t - \boldsymbol{w}^*\right\|^2 + \eta_t^2 c_1^2 L_{\max}^2\left\|\boldsymbol{w}^t - \boldsymbol{w}^*\right\|^2 \\
&= (1 - \mu_{i_t}\eta_t c_1 + \eta_t^2 L_{\max}^2 c_1^2)\left\|\boldsymbol{w}^t - \boldsymbol{w}^*\right\|^2.
\end{aligned}
\tag{18}
$$

Taking expectation *w.r.t.* $i_t$, we have that

$$
\begin{aligned}
\mathbb{E}\{\left\|\boldsymbol{w}^{t+1} - \boldsymbol{w}^*\right\|^2\} &\leq \mathbb{E}_{i_t}\{(1 - \mu_{i_t}\eta_t c_1 + \eta_t^2 L_{\max}^2 c_1^2)\left\|\boldsymbol{w}^t - \boldsymbol{w}^*\right\|^2\} \\
&= (1 - \mathbb{E}_{i_t}\{\mu_{i_t}\}\eta_t c_1 + \eta_t^2 L_{\max}^2 c_1^2)\left\|\boldsymbol{w}^t - \boldsymbol{w}^*\right\|^2 \\
&= (1 - \bar{\mu}\eta_t c_1 + \eta_t^2 L_{\max}^2 c_1^2)\left\|\boldsymbol{w}^t - \boldsymbol{w}^*\right\|^2.
\end{aligned}
\tag{19}
$$

Using recursion, we have that

$$\mathbb{E}\{\|\boldsymbol{w}^T - \boldsymbol{w}^*\|^2\} \leq (1 - \bar{\mu}\eta_t c_1 + \eta_t^2 c_1^2 L_{\max}^2)^T \|\boldsymbol{w}^0 - \boldsymbol{w}^*\|^2 \tag{20}$$

where we typically make $\eta_t c_1$ a constant such that $(1 - \bar{\mu}\eta_t c_1 + \eta_t^2 c_1^2 L_{\max}^2)$ is also a constant between $0$ and $1$. This is equivalent to the statement in the theorem that at most $\lceil (\log \frac{1}{1 - c_1 \eta_t \bar{\mu} + \eta_t^2 c_1^2 L_{\max}})^{-1} \log(\frac{1}{\epsilon}\|\boldsymbol{w}^0 - \boldsymbol{w}^*\|^2) \rceil$ iterations are needed to achieve the $\epsilon$-approximation $\mathbb{E}\{\|\boldsymbol{w}^T - \boldsymbol{w}^*\|^2\} \leq \epsilon$. $\qquad\square$

# C Proof of Theorem 2

*Proof.* The gradient update rule with the greedy LAST teacher is given by:

$$\boldsymbol{w}^{t+1} = \boldsymbol{w}^t - \eta_t \nabla_{\boldsymbol{w}^t} \ell(\boldsymbol{x}_{i_t}, y_{i_t} | \boldsymbol{w}^t) \tag{21}$$

where $y_{i_t}$ is the label synthesized by the teacher for $\boldsymbol{x}_{i_t}$. It can be rewritten as

$$\boldsymbol{w}^{t+1} = \boldsymbol{w}^t - \eta_t g(y_{i_t}) \nabla_{\boldsymbol{w}^t} \ell(\boldsymbol{x}_{i_t}, \tilde{y}_{i_t} | \boldsymbol{w}^t) \tag{22}$$

where $\tilde{y}_{i_t}$ is the ground truth label for $\boldsymbol{x}_{i_t}$. Therefore, we can simply define a new dynamic learning rate $\eta_t'(y_{i_t}) = \eta_t g(y_{i_t})$ and analyze the new gradient update rule based on the ground truth label:

$$\boldsymbol{w}^{t+1} = \boldsymbol{w}^t - \eta_t'(y_{i_t}) \nabla_{\boldsymbol{w}^t} \ell(\boldsymbol{x}_{i_t}, \tilde{y}_{i_t} | \boldsymbol{w}^t). \tag{23}$$

From the smoothness of $\ell_{i_t}$ and the new gradient update above, we have that

$$\ell_{i_t}(\boldsymbol{x}_{i_{t+1}}, \tilde{y}_{i_{t+1}} | \boldsymbol{w}^{t+1}) \leq \ell_{i_k}(\boldsymbol{x}_{i_t}, \tilde{y}_{i_t} | \boldsymbol{w}^t) - \left(\eta_t'(y_{i_t}) - \frac{L_{i_t}(\eta_t'(y_{i_t}))^2}{2}\right) \left\|\nabla \ell_{i_t}(\boldsymbol{x}_{i_t}, \tilde{y}_{i_t} | \boldsymbol{w}^t)\right\|^2. \tag{24}$$

Combining it with the condition:

$$\ell_{i_t}(\boldsymbol{x}_{i_{t+1}}, \tilde{y}_{i_{t+1}} | \boldsymbol{w}^{t+1}) \leq \ell_{i_k}(\boldsymbol{x}_{i_t}, \tilde{y}_{i_t} | \boldsymbol{w}^t) - c_2 \eta_t'(y_{i_t}) \left\|\nabla \ell_{i_t}(\boldsymbol{x}_{i_t}, \tilde{y}_{i_t} | \boldsymbol{w}^t)\right\|^2, \tag{25}$$

we will have that

$$c_2 \eta_t'(y_{i_t}) \geq \left(\eta_t'(y_{i_t}) - \frac{L_{i_t(\eta_t'(y_{i_t}))^2}}{2}\right) \tag{26}$$

which leads to $\eta_t'(y_{i_t}) \geq \frac{2(1-c_2)}{L_{i_t}}$. Considering that $\eta_t'(y_{i_t}) \leq \eta_{\max} := \max_t\{\eta_t\} \cdot \max_i\{g(y_i)\}$, we end up with

$$\eta_t'(y_{i_t}) \geq \min\left\{\frac{2(1-c_2)}{L_{i_t}}, \eta_{\max}'\right\}. \tag{27}$$

Then we consider the distance between $\boldsymbol{w}^{t+1}$ and $\boldsymbol{w}^*$:

$$\begin{aligned}
\left\|\boldsymbol{w}^{t+1} - \boldsymbol{w}^*\right\|^2 &= \left\|\boldsymbol{w}^t - \eta_t'(y_{i_t}) \nabla_{\boldsymbol{w}^t} \ell(\boldsymbol{x}_{i_t}, \tilde{y}_{i_t} | \boldsymbol{w}^t) - \boldsymbol{w}^*\right\|^2 \\
&= \left\|\boldsymbol{w}^t - \boldsymbol{w}^*\right\|^2 - 2\eta_t'(y_{i_t})\langle \nabla_{\boldsymbol{w}^t} \ell(\boldsymbol{x}_{i_t}, \tilde{y}_{i_t} | \boldsymbol{w}^t), \boldsymbol{w}^t - \boldsymbol{w}^*\rangle \\
&\quad + (\eta_t'(y_{i_t}))^2 \left\|\nabla_{\boldsymbol{w}^t} \ell(\boldsymbol{x}_{i_t}, \tilde{y}_{i_t} | \boldsymbol{w}^t)\right\|^2.
\end{aligned} \tag{28}$$

For the convenience of notation, we denote $\nabla_{\boldsymbol{w}^t} \ell(\boldsymbol{x}_{i_t}, \tilde{y}_{i_t} | \boldsymbol{w}^t)$ as $\nabla \ell_{i_t}(\boldsymbol{w}^t)$. Based on the strong convexity of $\ell_{i_t}(\cdot)$ (let $\mu_{i_t} = 0$ if the $\ell_{i_t}$ is not strongly convex), we have that

$$-\langle \nabla \ell_{i_t}(\boldsymbol{w}^t), \boldsymbol{w}^t - \boldsymbol{w}^*\rangle \leq \ell_{i_t}(\boldsymbol{w}^*) - \ell_{i_t}(\boldsymbol{w}^t) - \frac{\mu_{i_t}}{2}\left\|\boldsymbol{w}^t - \boldsymbol{w}^*\right\|^2. \tag{29}$$

Combining Eq. (28) and Eq. (29), we have that

$$\begin{aligned}
\left\|\boldsymbol{w}^{t+1} - \boldsymbol{w}^*\right\|^2 &\leq \left\|\boldsymbol{w}^t - \boldsymbol{w}^*\right\|^2 + 2\eta_t'(y_{i_t})\left(\ell_{i_t}(\boldsymbol{w}^*) - \ell_{i_t}(\boldsymbol{w}^t) - \frac{\mu_{i_t}}{2}\left\|\boldsymbol{w}^t - \boldsymbol{w}^*\right\|^2\right) \\
&\quad + (\eta_t'(y_{i_t}))^2 \left\|\nabla \ell_{i_t}(\boldsymbol{w}^t)\right\|^2 \\
&\leq (1 - \mu_{i_t}\eta_t'(y_{i_t}))\left\|\boldsymbol{w}^t - \boldsymbol{w}^*\right\|^2 + 2\eta_t'(y_{i_t})\left(\ell_{i_t}(\boldsymbol{w}^*) - \ell_{i_t}(\boldsymbol{w}^t)\right) \\
&\quad + \eta_t'(y_{i_t})\left\|\nabla \ell_{i_t}(\boldsymbol{w}^t)\right\|^2.
\end{aligned}$$

According to the condition that $\ell_{i_t}(\boldsymbol{w}^t - \eta_t g(y)\nabla \ell_{i_t}(\boldsymbol{w}^t)) \leq \ell_{i_t}(\boldsymbol{w}^t) - c_2 \eta_t g(y)\left\|\nabla \ell_{i_t}(\boldsymbol{w}^t)\right\|^2$, we have that

$$\begin{aligned}
\left\|\boldsymbol{w}^{t+1} - \boldsymbol{w}^*\right\|^2 &\leq (1 - \mu_{i_t}\eta_t'(y_{i_t}))\left\|\boldsymbol{w}^t - \boldsymbol{w}^*\right\|^2 + 2\eta_t'(y_{i_t})\left(\ell_{i_t}(\boldsymbol{w}^*) - \ell_{i_t}(\boldsymbol{w}^t)\right) \\
&\quad + \frac{\eta_t'(y_{i_t})}{c_2}\left(\ell_{i_t}(\boldsymbol{w}^t) - \ell_{i_t}(\boldsymbol{w}^{t+1})\right).
\end{aligned} \tag{30}$$

From the interpolation condition, we can obtain that $\boldsymbol{w}^*$ is the minimum for all functions $\ell_i$, which implies that $\ell_i(\boldsymbol{w}^*) \leq \ell_i(\boldsymbol{w}^{t+1})$ for all $i$. Putting this into the inequality, we have that

$$\begin{aligned}
\left\|\boldsymbol{w}^{t+1} - \boldsymbol{w}^*\right\|^2 &\leq (1 - \mu_{i_t}\eta_t'(y_{i_t}))\left\|\boldsymbol{w}^t - \boldsymbol{w}^*\right\|^2 + 2\eta_t'(y_{i_t})\left(\ell_{i_t}(\boldsymbol{w}^*) - \ell_{i_t}(\boldsymbol{w}^t)\right) \\
&\quad + \frac{\eta_t'(y_{i_t})}{c_2}\left(\ell_{i_t}(\boldsymbol{w}^t) - \ell_{i_t}(\boldsymbol{w}^*)\right)
\end{aligned} \tag{31}$$

which can be simplified to

$$\left\|\boldsymbol{w}^{t+1} - \boldsymbol{w}^*\right\|^2 \leq (1 - \mu_{i_t}\eta_t'(y_{i_t}))\left\|\boldsymbol{w}^t - \boldsymbol{w}^*\right\|^2 + \left(2\eta_t'(y_{i_t}) - \frac{\eta_t'(y_{i_t})}{c_2}\right)(\ell_{i_t}(\boldsymbol{w}^*) - \ell_{i_t}(\boldsymbol{w}^t)) \quad (32)$$

where the term $\left(\ell_{i_t}(\boldsymbol{w}^*) - \ell_{i_t}(\boldsymbol{w}^t)\right)$ is negative. We let $c_2 \geq \frac{1}{2}$, and therefore we will have that for for all $\eta_t'(y_{i_t})$,

$$\left(2\eta_t'(y_{i_t}) - \frac{\eta_t'(y_{i_t})}{c_2}\right) \geq 0 \quad (33)$$

which finally leads to

$$\left\|\boldsymbol{w}^{t+1} - \boldsymbol{w}^*\right\|^2 \leq (1 - \mu_{i_t}\eta_t'(y_{i_t}))\left\|\boldsymbol{w}^t - \boldsymbol{w}^*\right\|^2. \quad (34)$$

Then we take expectation *w.r.t.* $i_t$ on both sides:

$$\begin{aligned}
\mathbb{E}\{\left\|\boldsymbol{w}^{t+1} - \boldsymbol{w}^*\right\|^2\} &\leq \mathbb{E}_{i_t}\left\{(1 - \mu_{i_t}\eta_t'(y_{i_t}))\left\|\boldsymbol{w}^t - \boldsymbol{w}^*\right\|^2\right\} \\
&= (1 - \mathbb{E}_{i_t}\{\mu_{i_t}\eta_t'(y_{i_t})\})\left\|\boldsymbol{w}^t - \boldsymbol{w}^*\right\|^2 \\
&\leq \left(1 - \mathbb{E}_{i_t}\left\{\mu_{i_t}\min\left\{\frac{2(1-c_2)}{L_{i_t}}, \eta_{\max}'\right\}\right\}\right)\left\|\boldsymbol{w}^t - \boldsymbol{w}^*\right\|^2.
\end{aligned} \quad (35)$$

When $c_2 = \frac{1}{2}$, we have that

$$\mathbb{E}\{\left\|\boldsymbol{w}^{t+1} - \boldsymbol{w}^*\right\|^2\} \leq \left(1 - \mathbb{E}_{i_t}\left\{\mu_{i_t}\min\left\{\frac{1}{L_{i_t}}, \eta_{\max}'\right\}\right\}\right)\left\|\boldsymbol{w}^t - \boldsymbol{w}^*\right\|^2. \quad (36)$$

We can discuss this under two scenarios. First, when $\eta_{\max}' < \frac{1}{L_{\max}}$, we have that

$$\begin{aligned}
\mathbb{E}\{\left\|\boldsymbol{w}^{t+1} - \boldsymbol{w}^*\right\|^2\} &\leq (1 - \mathbb{E}_{i_t}\{\mu_{i_t}\eta_{\max}'\})\left\|\boldsymbol{w}^t - \boldsymbol{w}^*\right\|^2 \\
&= (1 - \mathbb{E}_{i_t}\{\mu_{i_t}\}\eta_{\max}')\left\|\boldsymbol{w}^t - \boldsymbol{w}^*\right\|^2 \\
&= (1 - \bar{\mu}\eta_{\max}')\left\|\boldsymbol{w}^t - \boldsymbol{w}^*\right\|^2
\end{aligned} \quad (37)$$

which leads to

$$\mathbb{E}\{\left\|\boldsymbol{w}^T - \boldsymbol{w}^*\right\|^2\} \leq (1 - \bar{\mu}\eta_{\max}')^T\left\|\boldsymbol{w}^0 - \boldsymbol{w}^*\right\|^2. \quad (38)$$

Second, when $\eta_{\max}' \geq \frac{1}{L_{\max}}$, we have that

$$\begin{aligned}
\mathbb{E}\{\left\|\boldsymbol{w}^{t+1} - \boldsymbol{w}^*\right\|^2\} &\leq \left(1 - \mathbb{E}_{i_t}\left\{\mu_{i_t}\min\left\{\frac{1}{L_{\max}}, \eta_{\max}'\right\}\right\}\right)\left\|\boldsymbol{w}^t - \boldsymbol{w}^*\right\|^2 \\
&= (1 - \frac{\bar{\mu}}{L_{\max}})\left\|\boldsymbol{w}^t - \boldsymbol{w}^*\right\|^2
\end{aligned} \quad (39)$$

which leads to

$$\mathbb{E}\{\left\|\boldsymbol{w}^T - \boldsymbol{w}^*\right\|^2\} \leq \left(1 - \frac{\bar{\mu}}{L_{\max}}\right)^T\left\|\boldsymbol{w}^0 - \boldsymbol{w}^*\right\|^2. \quad (40)$$

Combining both Eq. (38) and Eq. (40), we have that

$$\mathbb{E}\{\left\|\boldsymbol{w}^T - \boldsymbol{w}^*\right\|^2\} \leq \max\left\{\left(1 - \frac{\bar{\mu}}{L_{\max}}\right), (1 - \bar{\mu}\eta_{\max}')\right\}^T\left\|\boldsymbol{w}^0 - \boldsymbol{w}^*\right\|^2. \quad (41)$$

which concludes the proof. It implies that $c_3 = \left(\log\frac{1}{\max\left\{\left(1 - \frac{\bar{\mu}}{L_{\max}}\right), (1 - \bar{\mu}\eta_{\max}')\right\}}\right)^{-1}$ and $c_3\log(\frac{1}{\epsilon}\left\|\boldsymbol{w}^0 - \boldsymbol{w}^*\right\|^2)$ samples are needed to achieve $\mathbb{E}\{\left\|\boldsymbol{w}^T - \boldsymbol{w}^*\right\|^2\} \leq \epsilon$. $\qquad\square$

# D  Proof of Theorem 3

With $g(\boldsymbol{y}) = \eta_t^{-1} (\nabla_{\boldsymbol{w}}^2 f(\alpha \boldsymbol{w}^* + (1-\alpha)\boldsymbol{w}))^{-1}$, the gradient update becomes

$$\boldsymbol{w}^{t+1} = \boldsymbol{w}^t - (\nabla_{\boldsymbol{w}}^2 f(\alpha \boldsymbol{w}^* + (1-\alpha)\boldsymbol{w}^t))^{-1} \nabla_{\boldsymbol{w}} f(\boldsymbol{w}^t) \tag{42}$$

Because we are given that $|\lambda_{\min}(\nabla_{\boldsymbol{w}}^2 f(\boldsymbol{w}))| \geq \mu$, then we have for $\boldsymbol{w}$ in the $\sigma$-neighborhood that

$$\left\| (\nabla_{\boldsymbol{w}}^2 f(\boldsymbol{w}))^{-1} \right\| \leq \mu. \tag{43}$$

We write $\nabla_{\boldsymbol{w}}^2 f(\boldsymbol{w}^t)$ as the following equation:

$$\nabla_{\boldsymbol{w}} f(\boldsymbol{w}^t) = \int_0^1 \nabla_{\boldsymbol{w}}^2 f(\boldsymbol{w}^* + t(\boldsymbol{w}^t - \boldsymbol{w}^*))^\top (\boldsymbol{w}^t - \boldsymbol{w}^*) \mathrm{d}t. \tag{44}$$

Then we estimate the discrepancy between $\boldsymbol{w}^{t+1}$ and $\boldsymbol{w}^*$:

$$\left\| \boldsymbol{w}^{t+1} - \boldsymbol{w}^* \right\|$$
$$= \left\| \boldsymbol{w}^t - \boldsymbol{w}^* - (\nabla_{\boldsymbol{w}}^2 f(\alpha \boldsymbol{w}^* + (1-\alpha)\boldsymbol{w}^t))^{-1} \nabla_{\boldsymbol{w}} f(\boldsymbol{w}^t) \right\|$$
$$= \left\| (\nabla_{\boldsymbol{w}}^2 f(\alpha \boldsymbol{w}^* + (1-\alpha)\boldsymbol{w}^t))^{-1} \left( (\nabla_{\boldsymbol{w}}^2 f(\alpha \boldsymbol{w}^* + (1-\alpha)\boldsymbol{w}^t)) \cdot (\boldsymbol{w}^t - \boldsymbol{w}^*) - \nabla_{\boldsymbol{w}} f(\boldsymbol{w}^t) \right) \right\|$$
$$= \left\| (\nabla_{\boldsymbol{w}}^2 f(\alpha \boldsymbol{w}^* + (1-\alpha)\boldsymbol{w}^t))^{-1} \left( \nabla_{\boldsymbol{w}}^2 f(\alpha \boldsymbol{w}^* + (1-\alpha)\boldsymbol{w}^t) \right. \right.$$
$$\left. \left. - \int_0^1 \nabla^2 f(\boldsymbol{w}^* + t(\boldsymbol{w}^t - \boldsymbol{w}^*)) \mathrm{d}t \right)(\boldsymbol{w}^t - \boldsymbol{w}^*) \right\|$$
$$= \left\| (\nabla_{\boldsymbol{w}}^2 f(\alpha \boldsymbol{w}^* + (1-\alpha)\boldsymbol{w}^t))^{-1} \left( \int_0^1 (\nabla_{\boldsymbol{w}}^2 f(\alpha \boldsymbol{w}^* + (1-\alpha)\boldsymbol{w}^t) \right. \right.$$
$$\left. \left. - \nabla_{\boldsymbol{w}}^2 f(\boldsymbol{w}^* + t(\boldsymbol{w}^t - \boldsymbol{w}^*))) \mathrm{d}t \right)(\boldsymbol{w}^t - \boldsymbol{w}^*) \right\|$$
$$\leq \mu \left( \int_0^1 \left\| \nabla_{\boldsymbol{w}}^2 f(\alpha \boldsymbol{w}^* + (1-\alpha)\boldsymbol{w}^t) - \nabla_{\boldsymbol{w}}^2 f(\boldsymbol{w}^* + t(\boldsymbol{w}^t - \boldsymbol{w}^*)) \right\| \mathrm{d}t \right) \left\| \boldsymbol{w}^t - \boldsymbol{w}^* \right\|$$
$$= \mu \left( \int_0^1 L|1 - \alpha - t| \cdot \left\| \boldsymbol{w}^t - \boldsymbol{w}^* \right\| \mathrm{d}t \right) \left\| \boldsymbol{w}^t - \boldsymbol{w}^* \right\|^2$$
$$= \mu L \cdot \int_0^1 |1 - \alpha - t| \mathrm{d}t \cdot \left\| \boldsymbol{w}^t - \boldsymbol{w}^* \right\|^2$$
$$= \mu L \frac{(1-\alpha)^2 + \alpha^2}{2} \left\| \boldsymbol{w}^t - \boldsymbol{w}^* \right\|^2$$

When $\alpha = 0$ and $\alpha = 1$, we will have quadratic convergence (*i.e.*, super-exponential teachability) if $\frac{\mu L \delta}{2} < 1$. In general, we need $\frac{\mu L((1-\alpha)^2 + \alpha^2)\delta}{2} < 1$ to achieve super-ET. $\qquad \square$

# E Connection between Large-Margin Softmax and LAST

From [51], we consider the large-margin softmax loss (typically used for neural networks) as

$$
\begin{aligned}
L &= -\log\left(\frac{\exp(f_y)}{\sum_j \exp(f_j)}\right) \\
&= -\log\left(\frac{\exp(\|\boldsymbol{W}_y\|\|\boldsymbol{x}\|\psi(\theta_y))}{\exp(\|\boldsymbol{W}_y\|\|\boldsymbol{x}\|\psi(\theta_y)) + \sum_{j\neq y}\exp(\|\boldsymbol{W}_j\|\|\boldsymbol{x}\|\cos(\theta_j))}\right)
\end{aligned} \tag{45}
$$

where $f_j = \|\boldsymbol{W}_j\|\|\boldsymbol{x}\|\cos(\theta_j)$, $\boldsymbol{W}_j$ is the classifier of the $j$-th class, $y$ denotes the ground truth class, $\boldsymbol{x}$ is the features and $\theta_j$ is the angle between $\boldsymbol{x}$ and the $j$-th classifier $\boldsymbol{W}_j$. $\psi(\theta_y)$ is the key that differentiates large-margin softmax and the standard softmax. Typically to inject large margin between learned features, we need to let $\psi(\theta)$ to be always larger than $\cos(\theta)$ in the range of $[0, \pi]$.

It can be approximately viewed as teaching a special dynamic form of soft labels to the learner. Specifically, if we have a teacher model to generate a soft label $[a_1, a_2, \cdots, a_y, \cdots, a_K]$ where $K$ is the number of classes and $\sum_i a_i = 1$, we have the following cross-entropy loss:

$$
L = \sum_i -a_i \log\left(\frac{\exp(f_i)}{\sum_j \exp(f_j)}\right). \tag{46}
$$

According to Jensen's inequality, we have that

$$
\begin{aligned}
L &= \sum_i -a_i \log\left(\frac{\exp(f_i)}{\sum_j \exp(f_j)}\right) \\
&\geq -\log\left(\sum_i \frac{a_i \exp(f_i)}{\sum_j \exp(f_j)}\right).
\end{aligned} \tag{47}
$$

We can apply Jensen's inequality again and obtain that

$$
\begin{aligned}
-\log\left(\sum_i \frac{a_i \exp(f_i)}{\sum_j \exp(f_j)}\right) &\leq -\log\left(\frac{\exp(\sum_i a_i f_i)}{\sum_j \exp(f_j)}\right) \\
&= -\log\left(\frac{\exp(a_y f_y + \sum_{i\neq y} a_i f_i)}{\sum_j \exp(f_j)}\right) \\
&= -\log\left(\frac{\exp(a_y\|\boldsymbol{W}_y\|\|\boldsymbol{x}\|\cos(\theta_y) + \sum_{i\neq y} a_i\|\boldsymbol{W}_i\|\|\boldsymbol{x}\|\cos(\theta_i))}{\sum_j \exp(f_j)}\right) \\
&= -\log\left(\frac{\exp\left(\|\boldsymbol{W}_y\|\|\boldsymbol{x}\|(a_y\cos(\theta_y) + \sum_{i\neq y} b_i a_i \cos(\theta_i))\right)}{\sum_j \exp(f_j)}\right) \\
&= -\log\left(\frac{\exp\left(\|\boldsymbol{W}_y\|\|\boldsymbol{x}\|\psi(\theta_y)\right)}{\sum_j \exp(f_j)}\right)
\end{aligned}
$$

where we define that

$$
\psi(\theta_y) = a_y \cos(\theta_y) + \sum_{i\neq y} b_i a_i \cos(\theta_i) \tag{48}
$$

where $b_i = \frac{\|\boldsymbol{W}_i\|}{\|\boldsymbol{W}_y\|}$ Therefore, we have the following surrogate loss $L_s$ for the original cross-entropy loss with soft labels $L$:

$$
L_s = -\log\left(\frac{\exp\left(\|\boldsymbol{W}\|\|\boldsymbol{x}\|\psi(\theta_y)\right)}{\sum_j \exp(f_j)}\right) \tag{49}
$$

where $\psi(\cdot)$ here is a dynamic margin function that depends on the current prediction of the learner ($\theta_i, \forall i$) and the learner model ($\boldsymbol{W}_i, \forall i$). As long as we generate a $\psi(\cdot)$ that is always larger than $\cos(\cdot)$, then the large-margin effects can be achieved. Therefore, it is highly possible to use our LAST framework to generate soft labels that serve as a dynamic large-margin softmax loss. This kind of loss (with static margin) has diverse and useful applications [50, 81]. In contrast, our LAST framework can enable dynamic margin effect (based on the current learner), and could potentially be more effective in these applications.

# F  Additional Discussions and Results

## F.1  Experiments of Teaching Logistic Regression Learner on Gaussian Data

The results of teaching logistic regression learner to perform binary classification on Gasussian distributed cluster data are given in Fig. 13. The experimental details are given in Appendix F. We see that greedy LAST consistently converges faster than SGD, while LAST without constraints achieve significantly faster convergence than IMT. Moreover, we can observe that the mix teaching (LAST+IMT) achieves the best convergence. The experiments further validates our conclusion in the main paper.

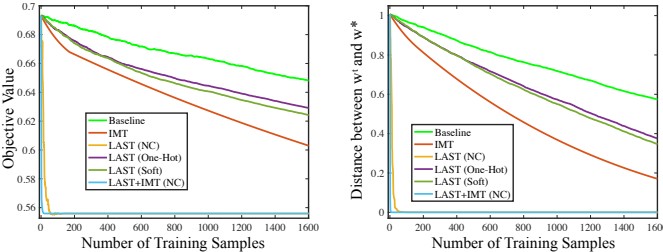

Figure 13: logistic regression learner on Gaussian data.

## F.2  Experiments of Teaching Logistic Regression Learner on Half-moon Data

We show the full convergence results in Fig. 14 for teaching logistic regression learner on half-moon data. The main paper shows some visualization results on synthesized labels, and we show the detailed objective values, distance between $w^t$ and $w^*$ and testing accuracy. The results again validate the effectiveness of greedy LAST.

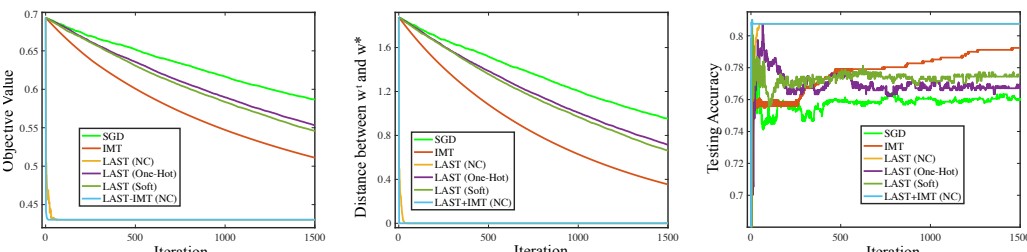

Figure 14: logistic regression learner on Gaussian data.

## F.3  Experiments of Teaching MLP Learner on MNIST

We show additional results on teaching MLP learners on MNIST in Fig. 15. Here we further present the 7/9 MNIST digit classification. We use greedy LAST to teach the MLP learners, and the settings are the same as the 3/5 digit classification in the main paper.

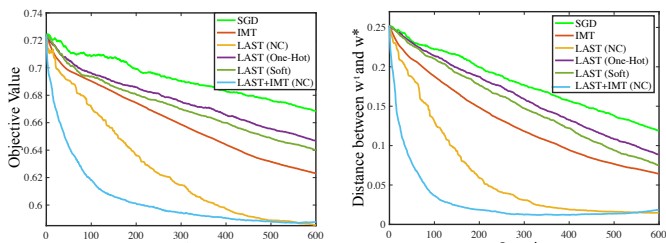

Figure 15: logistic regression learner on Gaussian data.

## F.4 Qualitative Results of LAST for Logistic Regression Learners

We show some teaching examples of greedy LAST in 3/5 classification. Fig. 16 shows that LAST tends to modify the labels of hard examples, while for easy examples, LAST will preserve the ground truth labels. LAST will typically synthesize the label of a hard example based on the current prediction so that the learner's convergence is less affected by the hard examples.

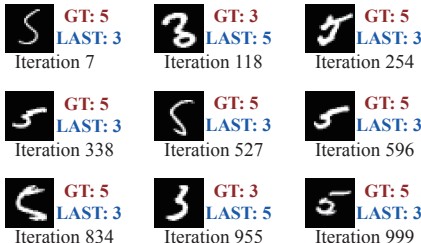

Figure 16: Some examples of labels generated by LAST (one-hot constraint).

## F.5 Omniscient Parameterized Teaching of Logistic Regression Learners

We also supplement the experiment of omniscient parameterize teaching (see Fig. 12 for the 1 batch size case) in the main paper with the case of 128 batch size. The other settings are exactly the same. Note that, for IMT, we still use the same setting where there is only one example selected for each iteration. The iteration-wise and time-wise convergence curves are given in Fig. 17.

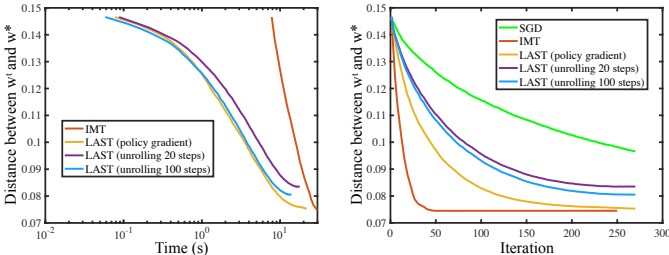

Figure 17: Omniscient parameterized teaching for logistic regression learners on binary classification on MNIST.

## F.6 Omniscient Parameterized Teaching of MLP Learners

We learn a parameterized teaching policy for 2-layer MLP learners. The teacher model is parameterized as a MLP. Details are given in Appendix G. We conduct multi-class classification of MNIST digit 1/2/3/4/5. For the unrolling, we unroll the teaching policy into 20 steps of gradient update. The results in Fig. 18 show that unrolling is powerful enough to learn an effective teaching policy that achieves faster convergence than SGD. We evaluate the batch size as 1 and 128 for both LAST and SGD. The convergence speed-up is very consistent.

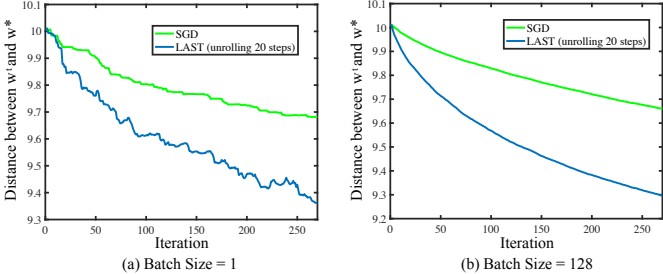

Figure 18: Omniscient parameterized teaching for MLP Learners on multi-class classification on MNIST digits.

### F.7 Effects of Different Action Spaces for LAST with Policy Gradient

On training omniscient parametrized teachers with reinforcement learning, we investigate the effects of scale in the action space. Our experimental setup is identical to that of Fig. 12 in the main paper. More specifically, we compare the action spaces for linear learners for binary classification on our projected MNIST dataset with otherwise identical settings in Appendix G:

- The default action space we use (augmented): {(0.0, 1.0), (0.25, 0.75), (0.5, 0.5), (0.75, 0.25), (1.0, 0.0), (0.0, 2.0), (0.5, 1.5), (1.0, 1.0), (1.5, 0.5), (2.0, 0.0)}.
- A strictly one-hot action space (non-augmented): {(0.0, 1.0), (0.25, 0.75), (0.5, 0.5), (0.75, 0.25), (1.0, 0.0)}.

We present the evaluation results with evaluation batch size 1 and 128 in Fig. 19. The results show that by having a more flexible and powerful action space, the teacher performance improves.

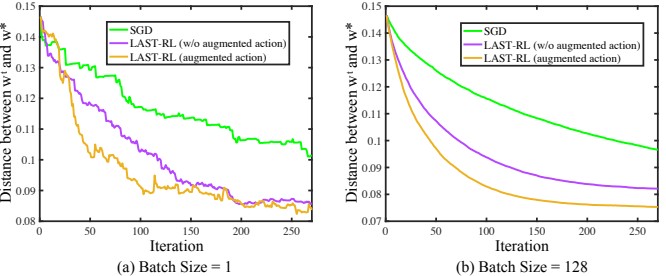

Figure 19: Effects of different action spaces for LAST with policy gradient.

### F.8 Black-box Parameterized Teaching of Logistic Regression Learners

To compare how unrolling and policy gradients perform in the black-box scenario, we first evaluate both unrolling and policy gradient on teaching a logistic regression learner on MNIST 3/5 digit classification. We parameterize the teacher with a MLP. Details are given in Appendix G. For the policy gradient, the space of synthesized label (*i.e.*, action space) includes $[0, 1]$, $[0.25, 0.75]$, $[0.5, 0.5]$, $[0.75, 0.25]$ and $[1, 0]$. For the unrolling, we unroll the teaching policy into 20 steps of the gradient update. The results in Fig. 20 demonstrate the great potential of BLAST to be able to consistently outperform SGD in the black-box scenario. Details of our settings are given in Appendix G.

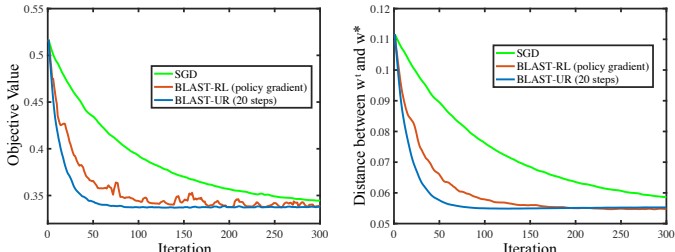

Figure 20: Black-box parameterized teaching for linear learners.

# G   Experimental Details

**General Settings.** For the experiments with linear models, the formulation of ridge regression is:

$$\min_{\boldsymbol{w}\in\mathbb{R}^d, b\in\mathbb{R}} \frac{1}{n}\sum_{i=1}^{n}\frac{1}{2}\left(\boldsymbol{w}^\top x_i + b - \hat{y}_i\right)^2 + \frac{\lambda}{2}\|\boldsymbol{w}\|^2 \tag{50}$$

where the synthesised label $\hat{y}_i$ is replaced by the true value $y_i$ in the SGD and IMT baselines, the constraint $r$ in LAST is defined as $r = \max_i \|y_i - \hat{y}_i\|_2$. The formulation of binary classification is:

$$\min_{\boldsymbol{w}\in\mathbb{R}^d, b\in\mathbb{R}} \frac{1}{n}\sum_{i=1}^{n}\mathcal{L}_{CE}\left(\hat{y}_i, \sigma_s\left(w^\top x_i + b\right)\right) + \frac{\lambda}{2}\|\boldsymbol{w}\|^2 \tag{51}$$

where $\mathcal{L}_{CE}(\cdot, \cdot)$ and $\sigma_s(\cdot)$ are the cross entropy loss and sigmoid function, respectively. We use the ground truth label $y_i$ instead of the synthesized counterpart $\hat{y}_i$ in the SGD and IMT baselines.

**Experiments on linear regression.** We first generate 800 data points $\boldsymbol{x}_{[i]} \in \mathbb{R}^4$, where $y_{[i]} = \langle \boldsymbol{w}^*, \boldsymbol{x}_i \rangle + b_i$, with additional Gaussian noise $0.02 \cdot \mathcal{N}(0, 1)$. For all methods, the learning rate is set as 0.001 and $\lambda$ is 0.00005.

**Experiments on synthetic data for classification.** For generating Gaussian cluster data, we use the 4-dimension data that is Gaussian distributed with $(0.2, \cdots, 0.2)$ (label +1) and $(-0.2, \cdots, -0.2)$ (label -1) as mean and identity matrix as covariance matrix. For half moon data, we use the API provided in scikit-learn, with 0.2 noise magnitude. For both synthesized data, we generate 400 training data points for each class, while the learning rate for the all methods is 0.0001, and $\lambda$ is set as 0.00005.

**Experiments on MNIST.** As in previous work [43], we use 24D random features (projected by a fixed and randomly generated matrix $\mathbb{R}^{784\times 24}$) of each image from the MNIST dataset. For non-parametrized teacher setting, the learning rate for all the compared methods are 0.001. $\lambda$ is set as 0.00005.

**Teacher architecture.** We use a two-hidden-layer MLP ('input dimension - 128 - 128 - output dimension') with ReLU activations for the whitebox and blackbox experiments on non-synthetic datasets unless otherwise specified.

**Omniscient teaching with parameterized teachers.** We create a subset of projected MNIST (24D random feature as before) by selecting 1000 images out of digit 3/5 for linear learner and digit 1/2/3/4/5 for MLP learner. The classifiers are randomly initialized around the optimal weights by adding a zero-mean Gaussian noise with standard deviation set to 5e-2 for linear learners and 1e-1 for MLP learners. The teacher simultaneously teaches 10 students (initialized differently) at the same time. We use an one-hidden-layer MLP architecture ('input dimension - 32 - number of classes') with a LeakyReLU activation (with coefficient 0.01) for the MLP learner. All learners do not use bias in the linear layers. We set learner update step size to 5e-4. During training, the batch that the learner receives is of size 1. For evaluation, the initialization and random seed is fixed. We try teaching batch size 1 and 128 for evalutation.

- *Unrolling.* We unroll the teaching process for $K$ steps. To make the teacher works well even after these steps, some students are reinitialized (with the same initialization method) after each $K$-step update and the remaining ones are not. For linear learners, we try unrolling 20 steps with 20% reset rate and 100 steps with 80% reset rate; for MLP learners, we show results of 20-step unrolling. We use Adam optimizer for the teacher with learning rate set to 1e-3, $(\beta_1, \beta_2) = (0.9, 0.999)$ and weight decay 1e-4. We train the model for 1000 episodes (each episode is a $K$-step inner update loop) for linear learners and 2000 episode for MLP learners. The objective for the teacher is a sum of whitebox L2 loss at each step, weighted by an exponential decay factor of 0.95 (similar in reinforcement learning but in reverse order). During testing, we fix the initialization of students and run 300 steps SGD with labels synthesized by the teacher. The input state is composed of 1) the input pair of images and labels, 2) the flattened and concatenated weight vectors and 3) the prediction of the learner given the input images.

- *Policy Gradient.* The state is composed of 1) the weight vectors (flattened and concatenated into one single vector) and 2) $M$ inner products between the displacement vector towards the optimal weights and the gradients given each possible action in a $M$-dim action space

where the actions are the synthesized labels. For binary classification, we consider this action space: $\{(0.0, 1.0), (0.25, 0.75), (0.5, 0.5), (0.75, 0.25), (1.0, 0.0), (0.0, 2.0), (0.5, 1.5), (1.0, 1.0), (1.5, 0.5), (2.0, 0.0)\}$. The reward at each step is the negative squared L2 distance to the optimal weights. We use the following variant of policy gradient:

$$\nabla J(\theta) \approx \frac{1}{N} \sum_{i=1}^{N} \sum_{t=1}^{T} \nabla_\theta \log \pi_\theta(a_t|s_t) \sum_{\tau=t}^{T} \gamma^\tau (r_\tau - b), \tag{52}$$

where $N$ is the number of trajectories, $T$ is the length of the episode, $\gamma$ is the discount rate and $b$ is the reward baseline for variance reduction. We set $T = 100$, $\gamma = 0.999$ and $b = -0.1$. $N = 10$ since we simultaneously train 10 students. Unlike the unrolling experiments, we randomly reset all students after each episode (with the same initialization method). We use the same Adam optimizer setting as in the unrolling experiments but without weight decay. We train the models until convergence and pick the best checkpoints along the trajectory.

**Blackbox teaching with parameterized teachers.** We set the teaching batch size to 20 for both training and evaluation, the step size for student updates to 1e-3 and use the student training cross-entropy loss (on a randomly sampled batch of size 20) instead of L2 distance to the optimal weights as the training objective. The other settings are kept the same as in whitebox teaching.