# OpenReview forum: "Iterative Teaching by Label Synthesis"
_NeurIPS.cc/2021/Conference — NeurIPS 2021 Spotlight_

### Official Review · Reviewer_7M1p · 2021-07-12

**Rating:** 7
**Confidence:** 4

**Summary:**

This paper studies the problem of iterative machine teaching with a focus on synthesizing the labels.

Specifically, the authors deal with the white-box (w* is known, the target parameter) and black-box teaching settings (w* is unknown). For the white-box setting, the authors propose a simple greedy teaching algorithm LAST, which is myopic. To incorporate the long-horizon information, the authors further present two methods: 1) unrolling the gradient steps and 2) learn a teaching policy with RL. These two methods are also applied to deal with the black-box setting.
For the theoretical analysis, the authors cast the effect of label synthesis/teaching as smartly tuning the learning rates, which also enjoys the exponential teachability under the convex and Lipschitz assumptions on the loss functions.

Empirical results show that the proposed methods are effective and require fewer iterations to converge than standard training (SGD).

**Limitations And Societal Impact:**

The authors discussed the societal impact and limitations in the experiments and concluding remarks.

**Main Review:**

This paper proposes three different algorithms for iterative machine teaching based on label synthesis. After reading the paper, I believe the main contribution is that this paper provides another angle to tackle the teaching problem with better computational efficiency compared to prior works. Overall, the proposed algorithms and the theoretical analysis are straightforward and incremental. In addition, the empirical results show that the proposed methods are more efficient than baselines, though there could be further improvements, e.g., the evaluation metric can be improved, the current one may not be suitable to support some of the claims. See detailed comments below.


1. Theorem 1 only bounds the expected difference between the learner's parameter w and the target parameter w*, rather than a worst-case bound or a probabilistic bounds. If the variance of the underlying distribution of w is large, then the bound on the expected difference is not informative. Therefore, I believe a probabilistic bound will be necessary.


2. In IMT [33], their bound is for the worst-case scenario, whereas your bound is for the average-case scenario. Therefore, I believe the argument "Theorem 1 is also stronger than the theoretical results in [33, 34],..." (Line235-236) is misleading.


3. Theorem 1 implies a linear convergence rate of LAST, which is the same for first-order optimization methods, such as stochastic gradient descent. Though the teaching monotonicity guarantees that LAST converges faster than SGD, it's still unclear how significant is the gain in terms of iterations? I believe some justifications on this point are essential, as LAST needs extra computational overheads for synthesizing the labels compared to vanilla SGD.


4. I think label synthesis is not relatively new and has been studied extensively in the literature. The following closely related works can serve as baselines in the experiments, e.g., Raghu et al. (2021) and Ren et al. (2018).  In particular, in your analysis, the effect of label synthesis can be regarded as tuning the learning rates for each teaching example, which is also equivalent to change the importance/weights of each sample. This perspective is related to Ren et al. (2018), where they also use the validation loss to learn the importance weight of each example by backpropagation (though for a different purpose). I believe it is the same idea as Eqn. (5) and Eqn. (6) in your paper.


5. A comparison in terms of the wall-clock time will be helpful. The comparisons using *iterations* are not fair to SGD as it has little overheads compared to your algorithms.


Questions:

6. Why is the cumulative reward not discounted by $\gamma$ for further steps in BLAST (Line 301)? However, for LAST, the reward is discounted (Line 204). Could you comment on this?


7. How do your methods compare to synthesizing the inputs?


8. In the introduction (Line 58), you mentioned "improving the generalization," how is this reflected in your analysis? I think this argument is not well-supported, especially for the white-box setting.

9. For parameterized teaching policy, is the teaching policy sensitive to different initializations, optimizers, learning rate and its schedule? Some ablation studies are necessary. In addition, it would be great to have one-standard error in the figure plot. (E.g., sample multiple w that have the same distance to w*, and the run your algorithms).


Overall, I think this paper attacks an exciting and essential problem. However, due to the above concerns on the theoretical results, the similarity of the proposed algorithms to prior works, and some of the arguments are not well-supported, I vote for weak rejection. I will increase my score if the authors can address my raised questions&concerns properly.

References:

[1] Raghu, Aniruddh, et al. "Teaching with commentaries." ICLR, 2021.

[2] Ren, Mengye, et al. "Learning to reweight examples for robust deep learning."  ICML, 2018

**Time Spent Reviewing:**

6

---

> ### Author Response · Authors · 2021-08-10
> **Response to Reviewer 7M1p (Part 2)**
>
> **Question 7: How do your methods compare to synthesizing the inputs?**
>
> **Response 7**: Very inspiring question! Synthesizing the inputs is in fact a general form of selecting teaching samples. In other words, selecting teaching samples can be viewed as a special case of input synthesis. Even sample selection is unscalable for large datasets. The biggest problem, as one might imagine, is the computational limitation. Synthesizing high-dimensional inputs is computationally prohibitive in practice (e.g. images typically have more than 1000 dimensions). Apart from the computational limitation, input synthesis (e.g. image generation) is by nature a more difficult problem and the learning space for the teaching policy is huge. Therefore, it may be very difficult to learn a good input synthesis policy.
>
> Nonetheless, we think it is a very exciting direction to explore. We are curious about what the input images will look like if they can speed up learner’s convergence. Even if we just learn a teaching policy that can transform / perturb the input image, it is already super interesting. It may give us many more interesting insights about data augmentation. It also connects well to the “learning to augment” series of works [d,e].
>
> [d] Zoph et al. Learning data augmentation strategies for object detection. ECCV 2020
>
> [e] Cubuk et al. Randaugment: Practical automated data augmentation with a reduced search space. CVPR 2020
>
> **Question 8: In the introduction (Line 58), you mentioned "improving the generalization," how is this reflected in your analysis? I think this argument is not well-supported, especially for the white-box setting.**
>
> **Response 8**: Sorry for the confusion. The improved generalization indeed can not be reflected in the analysis. One of the reasons is that our analysis is based on white-box iterative teaching where the goal is simply to guide the learner to the target parameters instead of improving generalization. This statement is mostly for the black-box teaching where there are no explicit target learner parameters. For black-box teaching, improved generalization can only be reflected empirically in experiments. We will modify the claim and make it more clear to avoid potential confusion in the revision.
>
> **Question 9: For parameterized teaching policy, is the teaching policy sensitive to different initializations, optimizers, learning rate and its schedule? Some ablation studies are necessary. In addition, it would be great to have one-standard error in the figure plot. (E.g., sample multiple w that have the same distance to w^star, and the run your algorithms).**
>
> **Response 9**: The sensitivity of the teaching policy is fully determined by the training settings. For the learner’s initialization, if we always stick to the same initialization for the learner during training, then the learned teaching policy is likely to be sensitive to initialization since it never sees another initialization during training. If we regularly change the initialization for the learner during training, then the teaching policy will no longer be sensitive to initialization. Empirically, we find that using the same learner initialization during training will learn a teaching policy more efficiently, while regularly changing the learner initialization during training will take longer for the teaching policy to converge. This can be viewed as data augmentation in training. Since learning rate and optimizer are typically fixed in practice, we typically fix them in training. However, it is also possible to perturbate them in training as additional data augmentation, if we want to make the learned teaching policy robust to learning rate and optimizer.
>
> Great suggestion! We will run the experiments with different initializations (by sampling from the same hypersphere whose center is ${w}^*$) and include the standard errors in the revision.

---

> > ### Comment · Reviewer_7M1p · 2021-08-18
> > **Additional questions**
> >
> > Thank you for the responses (also, the references) to address my concerns. Meanwhile, I have some additional questions:
> >
> > - Regarding the proof of theorem 1, one of the step (equation (9), Appendix) relies on
> > $$g(y)\cdot \nabla_w\ell(x,\tilde{y}|w)  = \nabla_w \ell(x,y|w),$$
> > where $g(y) = c_1 \cdot ||w-w^*||$. If I understand correctly, the generated label $y$ depends on the choice of $c_1$? For example, for LSR learner, the generated label y is obtained by solving the following equation,
> > $$\frac{\langle w, x \rangle - y}{\langle w, x \rangle -\tilde{y}} = c_1 ||w-w^*||. $$
> > My question is how does the above connects with the formulation of LAST? LAST generates the label by minimizing equation (1).
> > If label $y$ is generated by minimizing equation (1), then I don't think we can find a universal constant $c_1$ to ensure $g(y)=c_1 \cdot ||w-w^*||$.
> >
> >
> >
> > - Regarding the teaching monotonicity, the proof relies on the assumption that $\mathbb{E}[G(x, y|w_1)] \leq  \mathbb{E} [G(x, y|w_2)]$ for any $w_1, w_2$ that satisfy $\mathbb{E}[||w_1-w^*||]\leq  \mathbb{E} [||w_2-w^*||]$. This implies a property of diminishing returns of all the examples $(x, y)$. I think such property is hard to be satisfied in practice, e.g., neural network training. Also, without the teaching monotonicity, there is no guarantee that LAST will perform better than a random teacher.
> >
> >
> > - I also felt the paper is dense (this was also pointed out by other reviewers). I enjoyed reading the white-box teaching setting. The black-box setting is not that interesting, since the ideas are purely heuristic and also not new. So, I would suggest the authors move the black-box setting to the appendix.

---

> > > ### Author Response · Authors · 2021-08-19
> > > **Response to Additional Questions**
> > >
> > > Thanks a lot for the additional questions. We are more than happy to address them. We sincerely hope that our response can resolve your concerns. Any follow-up questions are welcome.
> > >
> > > **Question 1: the connection between the theory and greedy LAST**
> > >
> > > **Response 1**: Great question! First of all, Theorem 1 provides a lower bound of the average-case convergence rate for greedy LAST, which implies that greedy LAST can converge no slower than the rate shown in Theorem 1. Because it is less convenient to type formulae in openreview, we put the detailed derivation in the anonymous link.
> > >
> > > [Greedy LAST derivation]
> > > https://github.com/anonymous-neurips-submission/Anonymous-experiments/blob/main/greedy_LAST_proof.pdf
> > >
> > > Based on the derivation above, a universal constant $c_1$ is not necessary.
> > >
> > > We are so sorry that we jumped one step in the original proof of Theorem 1, which makes the connection to greedy LAST less obvious. We will add these derivations back to the proof.
> > >
> > > The general LAST framework is to solve $\min d(w^T,w^*)$ where $T$ is the termination step. Greedy LAST is a one-step approximation to solve this problem, where in each iteration, we only look ahead for one iteration and minimize the difference to $w^*$. In fact, we have also considered looking ahead multiple steps in the paper and it requires a parameterized teaching policy to make this practical. This is where the parameterized LAST comes from. Therefore, LAST is a general framework where greedy LAST and parameterized LAST represent different ways to solve the problem of guiding the learner to some target parameters.
> > >
> > > In this context, our convergence guarantees are very general and are not limited to the greedy LAST. The guarantee serves as a lower bound for the general LAST framework as well. The goal of the general LAST framework is to study the convergence gain brought by the teacher’s knowledge (e.g. the target model) through only synthesizing labels. In other words, the theoretical goal of LAST is to study the performance of some “oracle (optimal) teaching policy”. Our theoretical results can be viewed as a construction-based proof for the convergence lower bound of such an oracle teaching policy. In order to better understand iterative machine teaching, we are also interested in improving this lower bound and even finding an upper bound in the future.
> > >
> > >
> > > **Question 2: teaching monotonicity**
> > >
> > > **Response 2**: Thanks for the comments. We agree with the reviewer that for neural network training, the teaching monotonicity is indeed hard to satisfy in practice. However for linear learners, there are quite a few cases that can satisfy this property, e.g., least square regression learner. This has been shown in [a].
> > >
> > > One important thing to clarify: without teaching monotonicity, the LAST teacher still  converges faster than a random teacher (i.e. SGD) in terms of the convergence rate. This is already shown by Theorem 1.
> > >
> > > Teaching monotonicity is used to evaluate another aspect of the convergence (other than the convergence rate). In some sense, teaching monotonicity can be viewed as evaluating the stability of the convergence gain from the LAST teacher. For example, given the same initialization and the same random seed for sampling examples, LAST can **always** converge faster than a random teacher in **every** iteration with teaching monotonicity. This claim cannot be guaranteed by having a better convergence rate, since it is technically possible for a random teacher to converge faster than the LAST teacher with some “lucky” random seeds. Moreover, a better convergence rate also does not guarantee faster convergence in arbitrary iteration. However, with teaching monotonicity, as long as the random teacher and the LAST teacher share the same random seed, then the LAST teacher converges faster than the random teacher in every iteration. If the random teacher and the LAST teacher do not share the same random seed, teaching monotonicity can still guarantee that the LAST teacher converges faster than the random teacher in every iteration in an expectation sense.
> > >
> > > [a] Liu et al. Iterative Machine Teaching. ICML 2017
> > >
> > > **Question 3: paper structure**
> > >
> > > **Response 3**: Thanks for the great suggestion! We will adjust the paper structure in the revision to enrich the content and discussion in white-box teaching and move the black-box teaching to the appendix.

---

> > > > ### Comment · Reviewer_7M1p · 2021-08-19
> > > > **Thanks for the response**
> > > >
> > > > Thanks for your detailed response. I am happy to increase my rating from 5 to 7. One minor suggestion: it might be helpful to highlight the difference in convergence rates of LAST and SGD in the main paragraph.

---

> > > > > ### Author Response · Authors · 2021-08-19
> > > > > **Response**
> > > > >
> > > > > Thanks a lot for the suggestion! We will do so in the revision.

---

> ### Author Response · Authors · 2021-08-10
> **Response to Reviewer 7M1p (Part 1)**
>
> Thanks for many constructive comments. We are deeply appreciative of the reviewer’s efforts to help us improve our paper. We take all comments seriously and try our best to address every raised concern. We sincerely hope that our response can resolve your concerns. Any follow-up questions are welcome.
>
>
> **Question 1: Theorem 1 only bounds the expected difference between the learner's parameter w and the target parameter w^star, rather than a worst-case bound or a probabilistic bounds. If the variance of the underlying distribution of w is large, then the bound on the expected difference is not informative. Therefore, I believe a probabilistic bound will be necessary.**
>
> **Response 1**: Great suggestion! We do believe a probabilistic bound can be helpful, but the current theorems are still informative. The variance comes from the uniform sampling procedure which is the same for both SGD and LAST. Therefore, it is highly unlikely that the variance will be dramatically different for SGD and LAST. In such a sense, comparing the convergence rate in expectation for SGD and LAST is still meaningful.
>
> In fact, the expectation bounds are standard results when we discuss convergence rate for any stochastic optimization procedure. In contrast, high probability bounds are actually unusual even for SGD, and they typically require more involved analysis. There are only a few papers that obtain high probability bounds for classic SGD [a,b].
>
> To address the reviewer’s concerns, we still try our best to give a high probability bound. Note that this bound is by no means optimal for LAST, but due to the time constraint, this is the best we can get for now. We re-use the results of [a] in our setting and set $g(y)=\frac{2}{\eta_t \mu(t+1)}$. With a small change of conditions, we can have the high probability bound of $\mathcal{O}(1/t)$ rate. Although SGD can achieve the same high probability bound, our condition is actually more relaxed. For standard SGD to achieve this rate, the step size $\eta_t$ has to strictly be $\frac{2}{\mu(t+1)}$. In LAST, any step size can still achieve the same high probability rate, because of the presence of the teacher in LAST. This actually further demonstrates the superiority of LAST. Moreover, we still have the teaching monotonicity to guarantee that LAST can always be better than SGD.
>
> We will improve this high probability bound and discuss these results in the revision.
>
> [a] Harvey et al. Simple and optimal high-probability bounds for strongly-convex stochastic gradient descent. arXiv:1909.00843
>
> [b] Harvey et al. Tight analyses for non-smooth stochastic gradient descent. COLT 2019
>
> **Question 2: In IMT [33], their bound is for the worst-case scenario, whereas your bound is for the average-case scenario. Therefore, I believe the argument "Theorem 1 is also stronger than the theoretical results in [33, 34],..." (Line235-236) is misleading.**
>
> **Response 2**: Thanks for pointing it out. The convergence rate for LAST is in expectation due to the sampling procedure. In contrast, IMT has no stochastic component and everything is deterministic. However, it is also true that LAST requires a far less powerful teacher to achieve the same rate. We will fix the presentation to avoid any misleading claim in the revision.
>
> **Question 3: Theorem 1 implies a linear convergence rate of LAST, which is the same for first-order optimization methods, such as stochastic gradient descent. Though the teaching monotonicity guarantees that LAST converges faster than SGD, it's still unclear how significant is the gain in terms of iterations? I believe some justifications on this point are essential, as LAST needs extra computational overheads for synthesizing the labels compared to vanilla SGD.**
>
> **Response 3**: Typically for a loss that is strongly convex and has a Lipschitz gradient, SGD with decreasing step size can only achieve sub-linear convergence (i.e. $\mathcal{O}(1/t)$) [c]. For SGD to achieve linear convergence, there will be some additional costly procedures (such as line search). Vanilla SGD (what we use in the paper) can not achieve linear convergence. Therefore, in terms of iteration-wise convergence gain, it is linear convergence (LAST) vs. sub-linear convergence (SGD).
>
> LAST indeed needs extra overheads to synthesize labels, but it is not concerned with the iteration-wise convergence gain. For the time-wise convergence, we will empirically compare them later (we put the result in Question 5). We note that the iteration-wise acceleration is actually the focus of iterative machine teaching, as iterative teaching studies how a teacher can speed up the learning of a student via interactive constrained communication.
>
> We agree with the reviewer that some justifications on this point will be useful. We will add more discussion on the iteration-wise convergence gain in the revision.
>
> [c] Nemirovski et al. Robust stochastic approximation approach to stochastic programming, SIAM Journal on optimization, 2009
>
> **Question 4: I think label synthesis is not relatively new and has been studied extensively in the literature. The following closely related works can serve as baselines in the experiments, e.g., Raghu et al. (2021) and Ren et al. (2018). In particular, in your analysis, the effect of label synthesis can be regarded as tuning the learning rates for each teaching example, which is also equivalent to change the importance/weights of each sample. This perspective is related to Ren et al. (2018), where they also use the validation loss to learn the importance weight of each example by backpropagation (though for a different purpose). I believe it is the same idea as Eqn. (5) and Eqn. (6) in your paper.**
>
> **Response 4**: We have to clarify that label synthesis is by no means equivalent to weighting different samples. The equivalence approximately holds if the label is a scalar, which is almost never the case for applications such as image classification. Even in the case where the label is a scalar, LAST can be more flexible than sample weighting, because $g(y)$ can be negative while Ren et al. (2018) only considers the non-negative weighting. When the label is a vector, label synthesis will be much more flexible than changing the weights for different samples. If we have to draw a connection between label synthesis and sample weighting, we think it is safer to say label synthesis is a generalization for sample weighting (or sample weighting is a special case of label synthesis).
>
> What makes label synthesis particularly interesting and general to study is that it effectively unifies a number of popular methods such as sample weighting, label smoothing, label perturbation, knowledge distillation, self-training, co-training and many variants of softmax cross-entropy loss.
>
> We totally agree with the reviewer that both Raghu et al. (2021) and Ren et al. (2018) are highly related works and should be discussed, but we are attacking different problems than these works. The difference is mostly in two folds: (1) label synthesis vs. sample weighting; (2) teaching (where the teacher could have the access to the target parameters) vs. standard training (which is a special case of black-box teaching). We will discuss the connections and differences to these works in the revision.
>
> **Question 5: A comparison in terms of the wall-clock time will be helpful. The comparisons using iterations are not fair to SGD as it has little overheads compared to your algorithms.**
>
> **Response 5**: Thanks for the suggestion. To address the reviewer’s concerns, we perform the experiments to compare the time-wise convergence between SGD, IMT and LAST. Although LAST achieves the best time-wise convergence, we would like to emphasize that the focus of iterative machine teaching is the iteration-wise convergence which reflects the effectiveness of teacher-learner interaction.
>
> The setting of the following experiment is exactly the same as Figure 8 in the main paper. SGD takes a few minutes to finish, while IMT typically takes 10X more time to finish the same number of iterations. Detailed time-wise comparison can be found in the following anonymous links. We find that in terms of time-wise convergence, LAST outperforms both SGD and IMT by a significant margin. In contrast, IMT is even worse than SGD. Although it does not decrease the value of IMT, the results further show the significance of LAST.
>
> [Time-wise Loss Convergence]
> https://github.com/anonymous-neurips-submission/Anonymous-experiments/blob/main/time_loss_convergence.pdf
>
> [Time-wise Distance Convergence]
> https://github.com/anonymous-neurips-submission/Anonymous-experiments/blob/main/time_distance_convergence.pdf
>
> **Question 6: Why is the cumulative reward not discounted by for further steps in BLAST (Line 301)? However, for LAST, the reward is discounted (Line 204). Could you comment on this?**
>
> **Response 6**: In fact, both the discounted and undiscounted versions work well in our setting, but we empirically observe that for BLAST the undiscounted version generalizes slightly better into the future. As the return for each episode is a weighted sum of validation accuracy in each time step (in discounted case), there is an intrinsic trade-off between the final accuracy and the convergence speed of the student at test time. A change in discount factor might result in a change in both the desired teaching strategy and the optimization landscape for neural policies, and it can be seen as an example of reward shaping. We will make a clearer explanation in our paper.
>
> As a side note, we emphasize that the specific design choices for the RL-based teaching are not the major scope of our paper. It is very likely that an advanced design of the RL algorithm (e.g. Actor-Critic, a better reward design) can improve the empirical results. However, exhaustively trying them out is out of the scope of this paper.

---

### Official Review · Reviewer_vxUV · 2021-07-16

**Rating:** 8
**Confidence:** 5

**Summary:**

This paper proposes to address the sequential machine teaching problem by synthesizing labels. The proposed method is termed LAST. LAST avoids the expensive data selection and still provably improves the convergence of the learner. Specifically, LAST considers two types of teacher models: greedy teacher and parameterized teacher. For the parameterized teacher, the paper proposes unrolling and policy gradients to learn it. The theoretical justification is thorough and convincing. The experiments are comprehensive and well designed. They cover almost every setting one can think of. The empirical performance shows sufficient gains to support the major argument that LAST can effectively improve convergence.

**Limitations And Societal Impact:**

See sections above.

**Main Review:**

This paper is an interesting and enjoyable read. I like the overall idea of label synthesis teaching and how the paper executes the idea in a rather comprehensive way. More importantly, I find this paper quite inspiring and I have also learned a few useful insights from this paper. I summarize the strengths and potential weaknesses below.

** Strengths

- The proposed teaching framework is novel and well motivated. The paper studies a number of interesting variants under this framework. I believe this may be one of the first works that comprehensively study how modifying labels can affect convergence (in the supervised learning context). And this framework could potentially bridge the gap between sequential machine teaching and practical applications, since data selection is obviously not scalable enough for modern large-scale datasets.

- In general, labels are typically treated as a static learning signal in supervised learning. I think the proposed teaching framework suggests making such a static learning signal to be dynamically dependent on data and the learner model. This is conceptually novel and convincing to me, since uniform [1] and random [2] perturbation of the label can contribute to generalization, why not learn a teacher model to do this perturbation automatically in a data-driven way. I am convinced by this conceptual motivation (as mentioned in the introduction of the paper).

- While the proposed greedy LAST algorithm is simple and straightforward, the interpretation in Eq.(3) is super interesting and inspiring. The finding essentially suggests that in the linear setting, the optimal synthesized label for greedy teaching is a linear interpolation between the predicted label and the optimal label. I think this finding makes intuitive sense but there could be much more going on behind. In practice, I think this also gives us some guidance for how we can augment our labels to improve convergence and generalization.

- The idea of parameterizing the teacher with a neural network is novel and appealing in practice, as it makes the teacher much more scalable in terms of inference and also easier to train. In order to learn such a parameterized teacher, the paper proposes to use either unrolling or policy gradients. I think both are natural and effective choices. Unrolling could be more sample-efficient when the number of unrolling steps is sufficiently large.

- The paper also provides theoretical justifications of why LAST can achieve better convergence. They managed to achieve the same convergence rate as [2] by only modifying the labels, which seems to be a stronger result than [3].

- The experiments are well conducted and are very comprehensive. The LAST teaching framework generally shows consistently better convergence and generalization compared to the other methods. I like the fact that this paper uses quite a practice setting for sequential machine teaching, compared to prior work [3,4]. In the case of black-box teaching, the paper also compares a practical method L2T [5] (a RL-based data selection method), and shows better results. I believe this is quite encouraging and impressive.

- Figure 9 in the paper gives some empirical evidence and insights of how the synthesized labels look like. This is actually reflecting what the second point of strengths mentioned -- dynamic labels could potentially be better than static labels.

[1] Rethinking the Inception Architecture for Computer Vision, CVPR 2016
[2] DisturbLabel: Regularizing CNN on the Loss Layer, CVPR 2016
[3] Iterative Machine Teaching, ICML 2017
[4] An Optimal Control Approach to Sequential Machine Teaching, AISTATS 2019
[5] Learning to Teach, ICLR 2018


** Weaknesses

- Similar to most papers in machine teaching, the experimental setting is still far away from practical use and can only serve as a proof of concept. Although this paper already employs more practical experimental settings and relatively realistic datasets, the gap could be still further minimized.

- It will be more informative to test the convergence with respect to time in order to show the *real* convergence that people will care about in practice.

- I think unrolling generally takes a huge amount of GPU memory, which needs more discussions and justifications in the paper.

- The related work section can be improved by giving a more detailed discussion and comparison between LAST and the existing work.

- The paper is written in a dense manner and proposes a number of things. This is by no means a critical weakness, but making the paper too dense could distract the reader from understanding the essential idea of the paper. I would suggest authors try to restructure the content a bit. Maybe put some of the less important content to the appendix. Some of the content in the current appendix seems quite interesting (e.g. the connection to Armijo line search) and may be worth taking to the main paper.


** Additional Suggestions

- This paper actually raises a very interesting question “what are the best possible labels for supervised learning?”. I believe constructing dynamic data-model-dependent labels is actually a promising direction to explore. Beyond this, I think the label teaching is closely connected to information theory (mutual information, coding theory and data compression). Although I don’t have a clear picture of how these things are connected, it is obvious that a learner model tries to bridge the information encoded in the input and the labels. I suggest the authors dig deeper into this.

- The connection to privileged information is very interesting but not too obvious to me. I think the authors can elaborate more on this.

**Time Spent Reviewing:**

25

---

> ### Author Response · Authors · 2021-08-10
> **Response to Reviewer vxUV**
>
> Thanks for the encouraging comments and many constructive suggestions. We are deeply appreciative for the reviewer’s efforts to improve our paper. We take all comments seriously and try our best to address every raised concern. We sincerely hope that our response can address your concerns. Any follow-up questions are welcome.
>
>
> **Question 1: Similar to most papers in machine teaching, the experimental setting is still far away from practical use and can only serve as a proof of concept. Although this paper already employs more practical experimental settings and relatively realistic datasets, the gap could be still further minimized.**
>
> **Response 1**: Thanks for the suggestion. We agree that more large-scale datasets can strengthen our paper and make our results more convincing. Compared to the current machine teaching literature, our paper already adopts the most practical teaching scenarios and performs image classification experiments. In fact, we observed the same convergence gain from LAST on the CIFAR-10 dataset. Following the reviewer’s suggestion, we will add the CIFAR-10 results to the revised appendix.
>
> **Question 2: It will be more informative to test the convergence with respect to time in order to show the real convergence that people will care about in practice.**
>
> **Response 2**: Thanks for the suggestion. We have conducted the time-wise convergence experiment. The setting of the experiment is exactly the same as Figure 8 in the main paper. SGD takes a few minutes to finish, while IMT typically takes 10X more time. Detailed time-wise comparison can be found in the following anonymous links. We find that in terms of time-wise convergence, LAST outperforms both SGD and IMT by a significant margin. In contrast, IMT is even worse than SGD. These results further show the significance of LAST.
>
> [Time-wise Loss Convergence]
> https://github.com/anonymous-neurips-submission/Anonymous-experiments/blob/main/time_loss_convergence.pdf
>
> [Time-wise Distance Convergence]
> https://github.com/anonymous-neurips-submission/Anonymous-experiments/blob/main/time_distance_convergence.pdf
>
> **Question 3: I think unrolling generally takes a huge amount of GPU memory, which needs more discussions and justifications in the paper.**
>
> **Response 3**: Thanks for the suggestion. Unrolling more steps will indeed require more GPU memory. In our experiments, all the unrolled algorithms take less than 12GB GPU memory and fit in a single GPU. We will add the discussion about the GPU memory consumption in the revision.
>
> **Question 4: The related work section can be improved by giving a more detailed discussion and comparison between LAST and the existing work.**
>
> **Response 4**: Thanks for the suggestion. We totally agree with the reviewer that the related work section can be improved. We will substantially extend our current related work by adding more detailed discussion and comparison to the current machine teaching methods. Moreover, we will discuss the connection to soft-supervised learning and probabilistic cooperation / teaching.
>
> **Question 5: The paper is written in a dense manner and proposes a number of things. This is by no means a critical weakness, but making the paper too dense could distract the reader from understanding the essential idea of the paper. I would suggest authors try to restructure the content a bit. Maybe put some of the less important content to the appendix. Some of the content in the current appendix seems quite interesting (e.g. the connection to Armijo line search) and may be worth taking to the main paper.**
>
> **Response 5**: Great suggestion! We will spend more space to discuss motivation, related work and problem settings such that the content will be more accessible to a broader audience. We will also add more discussion about the connection to Armijo line search in the main paper.
>
> **Question 6: This paper actually raises a very interesting question “what are the best possible labels for supervised learning?”. I believe constructing dynamic data-model-dependent labels is actually a promising direction to explore. Beyond this, I think the label teaching is closely connected to information theory (mutual information, coding theory and data compression). Although I don’t have a clear picture of how these things are connected, it is obvious that a learner model tries to bridge the information encoded in the input and the labels. I suggest the authors dig deeper into this.**
>
> **Response 6**: Very interesting idea! We agree with the reviewer that learning dynamic learner-dependent labels is a promising direction to explore. As countless empirical evidences have suggested, the static ground truth labels are by no means the optimal supervision signals. White-box teaching aims to study this problem in a “supervised” fashion, because the target learner parameters are considered to be given. Black-box teaching aims to explore this in a more general and practical setting where there are no target parameters given. We believe that iterative teaching can provide many useful and effective insights in this direction. The connection to information theory is also super interesting to explore. Information theory may be useful to give an alternative interpretation to understand iterative teaching. It may also be very related to cooperative communication [a]. We will be interested to explore this in our future investigation.
>
> [a] Wang et al. A mathematical theory of cooperative communication, NeurIPS 2020
>
> **Question 7: The connection to privileged information is very interesting but not too obvious to me. I think the authors can elaborate more on this.**
>
> **Response 7**: Privileged information is defined as the additional information provided by an “intelligent teacher” for the student to better learn the target concept. As an example, the original data could be the image of a biopsy, and the privileged information is the medical report of an oncologist when inspecting the image, and the task is to predict a binary label indicating whether the tissue shown in the image is cancerous or healthy. See [b,c] for a comprehensive introduction. Typically, privileged information will induce better data separability and provably improve the convergence [c]. The connection to LAST is that the LAST teacher aims to implicitly encode the privileged information through labels. Different from the privileged information, the additional information in LAST comes from the knowledge of the target parameters ${w}^*$. In fact, the reviewer inspires us that privileged information can also be considered in the LAST framework. It will be super interesting to connect privileged information and LAST from a methodological perspective. We will add more discussion about the connection between privileged information and LAST in the revision.
>
> [b] Lopez-Paz et al. Unifying distillation and privileged information. ICLR 2016
>
> [c] Vapnik et al. Learning using privileged information: similarity control and knowledge transfer, JMLR 2015

---

> > ### Comment · Reviewer_vxUV · 2021-08-19
> > **Response to the author(s)**
> >
> > I am convinced by the response made by the author(s) to address my concerns. It is interesting that LAST achieves better time-wise efficiency than SGD, while IMT can barely do so. This is a significant advantage for LAST, and these results should be added to the revised paper. I trust the authors to also discuss those mentioned papers in their revised related work section.
> >
> > I also took some time to read the other reviews. The motivation of this paper does not seem to be a problem to me, although it can be improved and made more firendly for the broad audience (as suggested by Reviewer KxBd). From my understanding, the theoretical claims basically say that LAST can approximately have the same speed-up guarantees as the previous iterative teaching (or better speed-up with stronger assumptions as Thm 2 specifies). The average-case bound seems like a reasonable tradeoff when the stochasticity from data sampling is introduced to the teaching algorithm.
> >
> > Considering the potential broader impact of this paper, I will keep my current rating and vote for clear acceptance.

---

### Official Review · Reviewer_KxBd · 2021-07-19

**Rating:** 8
**Confidence:** 4

**Summary:**

The manuscript introduces LAST, an iterative approach to teaching by label synthesis. The authors develop an algorithmic framework, derive some theoretical results, and provide empirical analysis of the approach as compared to stochastic gradients, and prior iterative machine teaching methods based on example selection.

**Ethical Concerns:**

None.

**Limitations And Societal Impact:**

No limitations are acknowledged and societal impact is not discussed.

**Main Review:**

The paper presents a new approach to machine teaching based on label synthesis. The idea is simple and interesting: instead of selecting examples, which is a hard search problem through the database of examples, the authors propose to randomly select examples an optimize the labels. The resulting approach is flexible and general, as the authors demonstrate through consideration of different types of problems and models. Theoretical results show that the approach allows accelerated convergence as one would hope. Empirical results provide several demonstrations with interesting examples of when and why the approach would deviate from providing ground truth.

Strengths:
- (see above)
- Simple, interesting idea with a nice level of generality, effective theory, and empirical results.

Weaknesses:
- The approach is connected to a large number of literatures, most of which were not reviewed. Given space, it would be hard to do so. However, current discussion of related work omits several literatures mentioned by the authors. Other possible connections include soft supervised approaches and probabilistic versions of teaching / cooperation.
- The exposition is frustratingly vague at many points. Examples are supplied below. This is the largest weakness of the paper, which is that it is nowhere near self-encapsulated or even clear at many points. I would strongly implore the authors to adjust the exposition as the paper could be much more approachable for the broad audience who might find it interesting.

Specific questions and comments:
- "Then the student keeps learn22 ing from this batch dataset for the target concept." What does this mean?
- "[29] connects sequential teaching to optimal control and gains interesting insights, but it can not produce a practical teaching policy." This is not an adequate discussion of prior work. What constitutes interesting insights? What does "practical teaching policy" mean?
- There are several related literatures that are not discussed. Soft supervised learning is one that comes to mind. A second that is more closely related is the probabilistic literature on teaching and cooperation, e.g. Sequential cooperative bayesian inference.
- "Machine teaching is shown useful in reinforcement learning [53, 24, 45, 17], human-in the-loop learning [23, 6, 40], crowd sourcing [50, 67, 68] and cyber security [42, 2, 66, 64, 65]. [10, 14, 74, 47, 6, 26] study machine teaching from a more theoretical aspect." Please re-read this sentence.
- "We first consider the cleanest teaching protocol following" What does this mean? Why is it the cleanest? One should provide an explanation for a strong statement.
- "All these methods can be viewed as using a customized label synthesis policy in the black-box teaching scenario" It would be nice to have a more precise statement of this claim.
- Given that the approach unifies knowledge distillation, label smoothing, and self-training, it would be good to review at least some work in those literatures too.
- " This result validates our argument that it is not always optimal to feed the learner with ground truth label" I am not sure validates is the word you want here. Validation seems best provided by empirical demonstrations that it works on real data. "Is consistent with" may be a more suitable choice.
- The move from predicted to ground truth labels is interesting. I would be interested to hear the author's thoughts on probabilistic interpretations.
- " The predicted label is usually easy to learn for the current learner and the optimal label is more difficult to learn, implying that the learner should be taught in an easy-to-hard fashion." It would be helpful to explain why the predicted label is usually easy.
- "This implication nicely matches the conclusion in curriculum learning [5] and IM" What conclusion?
- " Moreover, greedy LAST also has the property of teaching monotonicity [33] if the learner loss satisfies certain conditions (Appendix C)." The conditions should be stated in the main text.
- Figure 3 didn't help me much. It would require more explaining in the caption than simply "example".
- " This nice property implies that LAST always converges to w∗ faster than a random teacher (i.e., SGD) if both of them sample teaching examples uniformly from the pool. An
illustration of how LAST works and why LAST yields better convergence to w∗ is given in Fig. 3. If the learner loss is properly designed, LAST can converge faster than SGD" These passages are repetitive and do not effectively explain.
- " The teaching monotonicity comes from the first iteration where the gradient update in LAST is guaranteed to better minimize the discrepancy between w1 and w∗." A more precise derivation/comparison would be helpful.
- " we propose a simple yet effective heuristic to teach MLP learners." How do we know this is an effective heuristic? A more detailed exposition here would be nice.
- " This shares similar spirits with back-propagation through time in recurrent networks and meta-learning" What does this mean? A more precise statement would be desirable.
- " The greedy policy is in fact the optimal solution to Eq. (5) for T = 1. For larger T, unrolling builds a larger computational graph for back-propagation and is more costly". It is helpful for the reader to provide a clear statement of why optimality is obtained. Also, given that computational considerations are one of the arguments in favor of LAST, an analysis of computational complexity here would clarify the argument.
- "Optimizing the unrolled teaching policy is conceptually similar to optimizing recurrent neural networks with back-propagation through time, as illustrated in Fig. 4. Unrolling T steps is equivalent to seeking a teaching policy that best minimizes the weight discrepancy after T-step gradient descent. Broadly speaking, our method also has intrinsic connections to learning an optimizer [4, 30] and teaching a loss function [61] in the sense that both aim to learn better gradients." Again, more precise statements would be helpful.
- "We can simply define rt =−kwt −w∗ k as the reward and directly apply policy gradient [59] to solve it." Please explain more.
- Theorem 1 contains several terms that are not defined.
- "Theorem 1 shows that g(y) plays a critical role, similar in spirit to the way that in line search, the step size is adaptively adjusted." Please re-read the sentence.
- "We connect Theorem 1 to Armijo linear search"  A more precise term than connect would be desirable.
- Figure 5 did not help me. The caption is inadequate.
- "Based on [33], it is easy to verify that mixed teaching can achieve ET" Please explain.
- "is able to achieve faster teaching empirically" please point to where this is demonstrated.
- The discussion of the data in figure 11 is interesting! (but not convincing) I would be interested to hear more about this.

**Time Spent Reviewing:**

4

---

> ### Author Response · Authors · 2021-08-10
> **Response to Reviewer KxBd (Part 3)**
>
> **Question 28: "is able to achieve faster teaching empirically" please point to where this is demonstrated.**
>
> **Response 28**: Sorry for not being clear enough. The empirical performance for the mixed teaching is given in Figure 7&8&10, labeled as “LAST+IMT”. We will fix this in the revision.
>
> **Question 29: The discussion of the data in figure 11 is interesting! (but not convincing) I would be interested to hear more about this.**
>
> **Response 29**: Figure 11 shows that LAST tends to modify the labels for semantically ambiguous examples which are usually seen as “hard examples”. We suspect this is because the separability of data theoretically affects the convergence (as shown in VC theory [p]). Modifying the labels of hard examples will likely improve the separability of data. Note that, the hardness of an example is also determined on the learner’s status (e.g., model parameters), so the teacher modifies the labels in a learner-dependent fashion. This implies that the label of the same example may be different across different learning stages.
>
> Moreover, the probabilistic interpretation may be particularly interesting here, since the samples with modified labels seem to present large uncertainty to the current learner. We will be super interested in looking into this in our future endeavour.
>
> [p] Vapnik et al. Learning using privileged information: similarity control and knowledge transfer, JMLR 2015

---

> ### Author Response · Authors · 2021-08-10
> **Response to Reviewer KxBd (Part 2)**
>
> **Question 14: " Moreover, greedy LAST also has the property of teaching monotonicity [33] if the learner loss satisfies certain conditions (Appendix C)." The conditions should be stated in the main text.**
>
> **Response 14**: Thanks for the suggestion. We will move the condition to the main paper in the revision.
>
> **Question 15: Figure 3 didn't help me much. It would require more explaining in the caption than simply "example".**
>
> **Response 15**: Sorry for not being clear enough. The figure essentially compares the vanilla SGD (i.e. random teacher) and label synthesis teacher. The label synthesis teacher will minimize the distance between current learner parameter ${w}^t$ and the target learner parameter ${w}^*$ along the same SGD gradient direction. We will change to a more informative caption for Figure 3 and add more explanations to the revision.
>
> **Question 16: " This nice property implies that LAST always converges to w∗ faster than a random teacher (i.e., SGD) if both of them sample teaching examples uniformly from the pool. An illustration of how LAST works and why LAST yields better convergence to w∗ is given in Fig. 3. If the learner loss is properly designed, LAST can converge faster than SGD" These passages are repetitive and do not effectively explain.**
>
> **Response 16**: These passages aim to convey that the label synthesis teacher can always converge faster than a random teacher (under suitable conditions). Figure 3 intuitively compares the random teacher and the label synthesis teacher. We will rewrite this part and add a suitable caption for Figure 3 for better clarity in the revision.
>
> **Question 17: " The teaching monotonicity comes from the first iteration where the gradient update in LAST is guaranteed to better minimize the discrepancy between w1 and w∗." A more precise derivation/comparison would be helpful.**
>
> **Response 17**: Thanks for the suggestion. For the same initialization and random seed (i.e., both SGD and LAST have the same gradient direction), LAST is guaranteed to be closer to the target learner parameter ${w}^*$ after the first update. Then after the first iteration, the teaching monotonicity takes over and guarantees that in the rest gradient updates, LAST always stays closer to ${w}^*$. After taking expectation, the statement holds in an expectation sense. We will move more formal statements in the appendix to the main paper in revision.
>
> **Question 18: " we propose a simple yet effective heuristic to teach MLP learners." How do we know this is an effective heuristic? A more detailed exposition here would be nice.**
>
> **Response 18**: Thanks for the suggestion. Theoretically, we have no guarantee for the teaching performance in a non-convex problem in general. This heuristic comes from a natural decomposition from the linear case, and we say it is effective because of its empirical performance. We have empirically tested it on MNIST and found it quite effective. We will modify this statement to be more precise in the revision.
>
> **Question 19: " This shares similar spirits with back-propagation through time in recurrent networks and meta-learning" What does this mean? A more precise statement would be desirable.**
>
> **Response 19**: Thanks for the suggestion. Unrolling the teacher model for $T$ iterations can be viewed as a recurrent neural network with $T$ timesteps. Back-propagation through time (BPTT) is a classic technique to optimize recurrent neural networks. BPTT unfolds a recurrent neural network in time. Every copy of the unfolded network shares the same parameters. Then the backpropagation algorithm is used to find the gradient of the cost with respect to all the network parameters. This strategy is also used in meta-learning [i]. We will add more descriptions and explanations in the revision.
>
> [i] Finn et al. Model-Agnostic Meta-Learning for Fast Adaptation of Deep Networks. ICML 2017
>
> **Question 20: " The greedy policy is in fact the optimal solution to Eq. (5) for T = 1. For larger T, unrolling builds a larger computational graph for back-propagation and is more costly". It is helpful for the reader to provide a clear statement of why optimality is obtained. Also, given that computational considerations are one of the arguments in favor of LAST, an analysis of computational complexity here would clarify the argument.**
>
> **Response 20**: Great suggestion! For the optimality of greedy policy, $T=1$ indicates that the teaching policy minimizes the discrepancy between one-step gradient update from the current learner parameters ${w}^t$ (i.e., ${w}^{t+1}$) and the target learner parameters ${w}^*$. This objective is exactly the same as the greedy policy, so the greedy policy is in fact the optimal teaching policy for the case of $T=1$.
>
> For the computational complexity, we give a brief demonstration from Line 32 to Line 37. But we agree with the reviewer that we should discuss this aspect more formally. We consider an example of teaching one sample in each iteration. Due to the sample selection over the entire dataset, the complexity of IMT is at least $\mathcal{O}(m)$ where $m$ is the size of the dataset. In contrast, the complexity of LAST is $\mathcal{O}(1)$ which is independent of the dataset size $m$. As the dataset size grows, the computational advantages of LAST will be increasingly beneficial.
>
> We will add the optimality discussion for the greedy policy and add a new section to discuss computational complexity to the revision.
>
> **Question 21: "Optimizing the unrolled teaching policy is conceptually similar to optimizing recurrent neural networks with back-propagation through time, as illustrated in Fig. 4. Unrolling T steps is equivalent to seeking a teaching policy that best minimizes the weight discrepancy after T-step gradient descent. Broadly speaking, our method also has intrinsic connections to learning an optimizer [4, 30] and teaching a loss function [61] in the sense that both aim to learn better gradients." Again, more precise statements would be helpful.**
>
> **Response 21**: Thanks for the suggestion. In iterative machine teaching, learning a good teacher model implies to learn a good gradient for the learner, which shares the same goal with “learning to learn” [j] and “learning to optimize” [k]. This is because all these works aim to produce a good gradient for a specific task to the learner. The difference mostly lies in the settings. Iterative machine teaching aims to study the convergence of the learner with the presence of a teacher that knows the target learner parameters. We will expand this discussion and add it to the revision.
>
> [j] Andrychowicz et al. Learning to learn by gradient descent by gradient descent, NeurIPS 2016
>
> [k] Li et al. Learning to optimize, arXiv:1606.01885
>
> **Question 22: "We can simply define rt =−kwt −w∗ k as the reward and directly apply policy gradient [59] to solve it." Please explain more.**
>
> **Response 22**: Thanks for the suggestion. Once we have defined the reward, state and action, we can learn the teacher by maximizing the reward with the classic REINFORCE algorithm (i.e., policy gradients) [m]. We will put all the implementation details in the appendix (current appendix should have sufficient details to reproduce our results, but in case we missed anything, we will double-check). Moreover, our code will also be published along with the paper.
>
> [m] Williams. Simple statistical gradient-following algorithms for connectionist reinforcement learning, Machine Learning, 1992
>
> **Question 23: Theorem 1 contains several terms that are not defined.**
>
> **Response 23**: Thanks for pointing it out. We will fix it in the revision.
>
> **Question 24: "Theorem 1 shows that g(y) plays a critical role, similar in spirit to the way that in line search, the step size is adaptively adjusted." Please re-read the sentence.**
>
> **Response 24**: Thanks for pointing it out. We will fix this sentence in the revision.
>
> **Question 25: "We connect Theorem 1 to Armijo linear search" A more precise term than connect would be desirable.**
>
> **Response 25**: Thanks for pointing it out. We will move some of the content in the appendix to the main paper and add more explanation for this connection in the revision.
>
> **Question 26: Figure 5 did not help me. The caption is inadequate.**
>
> **Response 26**: Sorry for not being clear enough. Figure 5 aims to show how mixed teaching compares to the original LAST. The major difference is the way the gradient direction is produced. LAST has the same initial gradient direction as SGD, while the mixed teaching has the same initial direction as IMT. We will add more description to the caption to ensure it can be understood easily.
>
> **Question 27: "Based on [33], it is easy to verify that mixed teaching can achieve ET" Please explain.**
>
> **Response 27**: The previous iterative teaching in [n] can achieve exponential teaching in the worst scenario, and this can immediately serve as the lower bound for the mixed teaching since we can construct the label exactly following [n]. We will add a formal statement and more explanations to improve the clarity in the revision.
>
> [n] Liu et al. Iterative machine teaching, ICML 2017

---

> ### Author Response · Authors · 2021-08-10
> **Response to Reviewer KxBd (Part 1)**
>
> Thanks for the encouraging comments and many constructive suggestions. We are deeply appreciative of the reviewer’s efforts to improve our paper. We take all comments seriously and try our best to address every raised concern. We sincerely hope that our response resolves your concerns. Any follow-up questions are welcome.
>
> **Question 1: Lack of discussion in the related work.**
>
> **Response 1**: Thanks for the suggestion. We will expand our related work section with more in-depth discussion with existing work. We agree with the reviewer that soft-supervised approaches (e.g., self-training, co-training) and probabilistic teaching (e.g., cooperative inference, Bayesian teaching) are highly related and should be discussed.
>
>
> **Question 2: Adjust the exposition as the paper could be much more approachable for the broad audience who might find it interesting.**
>
> **Response 2**: Great suggestion! We agree with the reviewer that it will make machine teaching more accessible for a broader audience by adjusting the current paper structure. Specifically, we will spend more space in the revision to discuss the problem settings, the motivation and the connection to existing works.
>
>
> **Question 3: "Then the student keeps learning from this batch dataset for the target concept." What does this mean?**
>
> **Response 3**: Sorry for the confusion. It refers to the setting for conventional batch-based machine teaching [a] where the teacher provides the dataset to the student in one shot and then the student uses its learning algorithm to learn from this static dataset for the optimal student parameters. We will improve this description for better clarity.
>
> [a] Zhu. Machine teaching: An inverse problem to machine learning and an approach toward optimal education, AAAI 2015
>
> **Question 4: "[29] connects sequential teaching to optimal control and gains interesting insights, but it can not produce a practical teaching policy." This is not an adequate discussion of prior work. What constitutes interesting insights? What does "practical teaching policy" mean?**
>
> **Response 4**: Sorry for the confusion. We say that this paper gives interesting insights because it connects sequential / iterative machine teaching to an optimal control problem. [b] provides a teaching trajectory rather than a teaching policy. A teaching policy can be viewed as a function where the input is the current status of the learner and the output is the teaching example (or synthesized labels in LAST). A teaching trajectory is a sequence of examples that guides a student from ${w}^0$ to ${w}^*$. Any numerical perturbation in the student model update (i.e., ${w}^t$) will require recomputing the trajectory.
>
> Most importantly, as one may expect, such a teaching trajectory is extremely difficult to compute in practice. It is only feasible in the case of a small dataset. This limitation is also discussed in the third paragraph of Section 5 in [b]. We quote from [b]: “A drawback of both NLP and CNLP is that they produce trajectories rather than policies”.
>
> We will substantially expand our current related work section to include a more in-depth review of existing works.
>
> [b] Lessard, Zhang, Zhu. An Optimal Control Approach to Sequential Machine Teaching, AISTATS 2019
>
> **Question 5: There are several related literatures that are not discussed. Soft supervised learning is one that comes to mind. A second that is more closely related is the probabilistic literature on teaching and cooperation, e.g. Sequential cooperative bayesian inference.**
>
> **Response 5**: Great suggestion! We have looked into both soft-supervised learning and probabilistic teaching / cooperation and found them extremely related to our work. The connection to soft-supervised learning [c] is natural, since it considers to relax the one-hot ground truth label to a soft label during learning.
>
> Particularly, we discover that the series of works in cooperation and probabilistic teaching (just to name a few papers that we found [d,e,f,g]) are extremely related to machine teaching in general. [d] considers a sequential teaching setting in a probabilistic way. [e] even considers a generic model to unify machine teaching and cooperative inference. We sincerely thank the reviewer for providing us with these important works and we will discuss the connection to them in the revision.
>
> [c] Subramanya, Bilmes. Soft-supervised learning for text classification, EMNLP 2018
>
> [d] Wang, et al. Sequential Cooperative Bayesian Inference, ICML 2020
>
> [e] Wang et al. A mathematical theory of cooperative communication, NeurIPS 2020
>
> [f] Yang et al. Optimal cooperative inference, AISTATS 2018
>
> [g] Eaves, Shafto. Toward a general, scalable framework for Bayesian teaching with applications to topic models, arXiv:1605.07999
>
> **Question 6: "Machine teaching is shown useful in reinforcement learning [53, 24, 45, 17], human-in the-loop learning [23, 6, 40], crowd sourcing [50, 67, 68] and cyber security [42, 2, 66, 64, 65]. [10, 14, 74, 47, 6, 26] study machine teaching from a more theoretical aspect." Please re-read this sentence.**
>
> **Response 6**: Thanks for pointing it out. We will further polish the presentation in the revision.
>
> **Question 7: "We first consider the cleanest teaching protocol following" What does this mean? Why is it the cleanest? One should provide an explanation for a strong statement.**
>
> **Response 7**: Sorry for the confusion. By “cleanest”, we actually mean “simplest”. We aim to start with a simplest iterative machine teaching setting where the teacher has access to all the information of the student. We will fix this in the revision.
>
> **Question 8: "All these methods can be viewed as using a customized label synthesis policy in the black-box teaching scenario" It would be nice to have a more precise statement of this claim.**
>
> **Response 8**: Thanks for the suggestion. Knowledge distillation, label smoothing and self-training are different heuristics to generate the labels for training data, so each of them corresponds to a specific black-box label synthesis policy (without knowing the status as well as the optimal parameters of the student). From Line 113 to Line 115, we distinguish our proposed method from these black-box teaching policies. We will add a more formal description and explanation in the revision.
>
>
> **Question 9: Given that the approach unifies knowledge distillation, label smoothing, and self-training, it would be good to review at least some work in those literatures too.**
>
> **Response 9**: Great suggestion! We will review these related works in the revision.
>
> **Question 10: " This result validates our argument that it is not always optimal to feed the learner with ground truth label" I am not sure validates is the word you want here. Validation seems best provided by empirical demonstrations that it works on real data. "Is consistent with" may be a more suitable choice.**
>
> **Response 10**: Thanks for the suggestion. We will fix it in the revision.
>
>
> **Question 11: The move from predicted to ground truth labels is interesting. I would be interested to hear the author's thoughts on probabilistic interpretations.**
>
> **Response 11**: It is super interesting to think about iterative machine teaching from a probabilistic viewpoint. We find that [h] is a good example of a probabilistic view for iterative teaching. There could be many interesting probabilistic interpretations. Broadly speaking, the balance between predicted labels and the ground truth labels may have corresponding probabilistic interpretations such as the balance between learner’s prior and the teacher’s likelihood / posterior. The result means that it is not always desirable to fully rely on the teacher if the learner’s goal is to achieve fast algorithmic convergence.
>
> We don’t have a very clear idea of what the probabilistic interpretation for Eq.(3) should be, but we do think it is an important open problem. There are two ways of attacking the iterative machine teaching problem in a probabilistic style. First, it might require us to reformulate the teaching settings where the target parameters become a probabilistic distribution. Then the iterative teaching may not use a distance to measure the learner discrepancy. Instead, the objective may be some divergence measure (e.g. KL). Second, we can also consider the teaching signal to be a label distribution (which allows the learner to sample) rather than a static soft label. It will greatly enrich the technical depth of iterative teaching. We will be super interested in investigating this in future work.
>
> [h] Wang, et al. Sequential Cooperative Bayesian Inference, ICML 2020
>
> **Question 12: " The predicted label is usually easy to learn for the current learner and the optimal label is more difficult to learn, implying that the learner should be taught in an easy-to-hard fashion." It would be helpful to explain why the predicted label is usually easy.**
>
> **Response 12**: Thanks for the suggestion. The predicted label is usually easy because the learner does not need to update itself in order to produce the predicted label, while the optimal label requires multiple rounds of updates for the learner. In other words, the more effort the learner needs to spend, the more difficult the label is to learn. We will add more explanation in the revision.
>
> **Question 13: "This implication nicely matches the conclusion in curriculum learning [5] and IM" What conclusion?**
>
> **Response 13**: Sorry for not being clear enough. Both curriculum learning and iterative machine teaching have reached the conclusion that typically feeding easy samples first and gradually shifting to difficult samples can improve the learner convergence. Curriculum learning [x] verifies this conclusion empirically, while iterative machine teaching shows this in a theoretical way. We will add more discussion in the revision.
>
> [x] Bengio et al. Curriculum learning. ICML 2009

---

> > ### Comment · Reviewer_KxBd · 2021-09-01
> > **Response to author response**
> >
> > Thanks to the authors for their reply! After reading the other reviews, and the author responses, and the responses to the responses, I am inclined to keep my rating where it is. Fun paper!

---

### Official Review · Reviewer_we7s · 2021-07-20

**Rating:** 6
**Confidence:** 2

**Summary:**

This paper proposes an algorithm for machine teaching that relies on synthesizing labels for randomly selected examples rather than selecting ideal examples.

**Limitations And Societal Impact:**

Yes.

**Main Review:**

Strengths
- Technique comes with theoretical guarantees

Weaknesses
- Unclear motivation
- Unclear meaningfulness of results

The problem of label synthesis is intriguing due to the effectiveness of using “soft labels” in practice. However, I’m not sure whether this paper provides useful answers to these questions.

In general, the results in this paper are not sufficiently motivated and explained. For instance, one key question I have is how this approach compares with a naive strategy that both randomly samples examples *and* uses the true labels (instead of the synthesized labels). This seems like a particularly important point of comparison. Under the assumptions made by the authors, how well does this approach perform? Is it provably worse than the proposed approach?

A related issue is that the paper generally does not do a good job of explaining how their results differ from the existing literature on machine teaching. For instance, are the convergence rates the same? Do they make the same assumptions about knowledge of the student loss function and optimal parameters?

Finally, another question I have is about the connection to active learning. There are many active learning algorithms that only make a single pass over the training data -- for instance, [A]. This appears to be applicable to the problem studied in this paper -- basically, only use examples selected by the active learning algorithm for teaching. How do the results in this paper compare to this strategy?

[A] Beygelzimer et al., Agnostic Active Learning Without Constraints. NeurIPS 2010.


**Time Spent Reviewing:**

1

---

> ### Author Response · Authors · 2021-08-10
> **Response to Reviewer we7s**
>
> Thanks for the useful comments. We sincrerely thank the reviewer's efforts for helping us improve the paper. With all respect, we are afraid that the reviewer may have some misunderstandings of our paper. We carefully address the reviewer’s concerns below and sincerely hope that our response resolves your concerns. Any follow-up questions are welcome.
>
> **Question 1: Unclear motivation**
>
> **Response 1**: Our motivation comes from (1) the computationally costly sample selection process in the previous iterative machine teaching literature, (2) the empirical effectiveness of non-ground-truth labels such as distilled labels in knowledge distillation and uniformly smoothed labels in label smoothing, and (3) the rich gradient space which becomes available by modifying labels. We will improve our presentation to reflect our motivation in a more clear way for a broader audience.
>
>
> **Question 2: Unclear meaningfulness of results**
>
> **Response 2**: Our theoretical results show that label synthesis teaching can achieve: 1) exponential teachability, and 2) even super-exponential teachability with additional assumptions. It shows that label synthesis can be sufficiently flexible for teaching. In white-box teaching, our empirical results show that label synthesis can outperform both the SGD and the previous iterative machine teaching methods by a considerable margin. In black-box teaching where the previous benchmarks are not applicable, our empirical results show that label synthesis can still be consistently better than the SGD. We will improve the experiment section to include more discussions.
>
>
> **Question 3: How this approach compares with a naive strategy that both randomly samples examples and uses the true labels (instead of the synthesized labels). Under the assumptions made by the authors, how well does this approach perform? Is it provably worse than the proposed approach?**
>
> **Response 3**: We have already performed this comparison for every experiment, please see the Figure 7, 8, 10, 12-15 and relevant discussions in the paper. In our experiments, the setting termed SGD is exactly what the reviewer describes -- training the learner with randomly sampled data with their ground truth labels. In terms of empirical performance, both LAST and BLAST consistently outperform this baseline. In terms of theoretical results, the convergence rate of SGD is known to be sub-linear, as opposed to the linear convergence (i.e. exponential teachability) achieved by LAST.
>
>
> **Question 4: A related issue is that the paper generally does not do a good job of explaining how their results differ from the existing literature on machine teaching. For instance, are the convergence rates the same? Do they make the same assumptions about knowledge of the student loss function and optimal parameters?**
>
> **Response 4**: Machine teaching can be divided into batch-based and iterative. Batch-based machine teaching focuses on the size of the teaching set, while iterative machine teaching focuses on learner convergence. From Line 15, we have clearly discussed the differences between these two machine teaching paradigms. For iterative machine teaching (IMT) [a,b], Figure 1 and the corresponding paragraphs have compared the differences between IMT and our method. The difference in convergence rate is also discussed in Line 236. We will highlight the comparison in the revision.
>
> Specifically, we have compared our results to the most related previous method -- IMT. Both IMT and our LAST can achieve exponential teachability, but LAST uses a less powerful teaching algorithm that can not select samples. All the analysis in IMT and LAST assumes a white-box teaching scenario, so they make the same assumptions on the student learner.
>
>
> **Question 5: There are many active learning algorithms that only make a single pass over the training data -- for instance, [A]. This appears to be applicable to the problem studied in this paper -- basically, only use examples selected by the active learning algorithm for teaching. How do the results in this paper compare to this strategy?**
>
> **Response 5**: Thanks for the suggestion. The active learner aims to achieve high accuracy using as few labeled instances as possible. This is achieved by finding suitable data to be labeled by an oracle. Active learning is closely related to batch-based machine teaching, since they share a similar goal to find a dataset with minimal size. Batch-based machine teaching has better sample complexity than active learning, because the teacher knows the optimal learner parameters. A detailed comparison has already been made in [c,d].
>
> However, for iterative machine teaching, active learning is not a suitable baseline, since it focuses on minimal labeled examples instead of algorithmic convergence. Moreover, active learning typically uses different metrics to measure its performance (see x-axis of Figure 2 in [A]), while iterative machine teaching uses the number of iterations. Therefore, it may not be meaningful to directly compare them.
>
> Inspired by the reviewer, we additionally implement one variant of LAST -- first using a uncertainty-based active learning to select a sample and then synthesizing labels for it. This can be viewed as using LAST as an oracle labeler. We are not sure whether this is the baseline requested by the reviewer, but we did it anyway. We empirically find that this variant performs worse than the vanilla LAST that uses uniform sampling. However, it may still be interesting to compare different active learning algorithms with LAST upon them (although it is out of the scope of this paper). We will include this comparison in the appendix in the revision.
>
>
> [a] Liu et al. Iterative machine teaching, ICML 2017
>
> [b] Liu et al. Towards black-box iterative machine teaching, ICML 2018
>
> [c] Zhu et al. An overview of machine teaching, arXiv preprint:1801.05927, 2018
>
> [d] Zhu. Machine teaching: An inverse problem to machine learning and an approach toward optimal education, AAAI 2015

---

> > ### Comment · Reviewer_we7s · 2021-09-01
> > **Response**
> >
> > Thank you for addressing my concerns; I will update my score accordingly. As a non-expert, I believe the exposition could be significantly improved to help with readability, especially in terms of motivation, and I hope the reviewers will clarify the points they discussed in their response.

---

### Decision · Program_Chairs · 2021-09-27

**Decision:**

Accept (Spotlight)

**Comment:**

This paper studied an interesting yet challenging problem in machine teaching and provided an intuitive teaching algorithm for synthesizing labels for teaching with theoretical analysis and justifications. All reviewers agree that the problem studied in this paper is practically relevant, with rigorous theoretical analysis and justifications. Empirical results on several large data sets seem promising.

Meanwhile, there were common concerns in the positioning of the work, in particular in the lack of discussion of connections to related work, and some reviews concern the clarity of the presentation (e.g. missing some details in experimental results such as computation time), and the significance of the theoretical contribution (e.g. whether the average-case bound is significant for practical scenarios). The authors provided effective feedback during the discussion phase, which helped clarify many of the above concerns. Should these concerns be addressed in the revision, it would make a solid paper. The authors are encouraged to take into account the feedback from the discussion phase to further improve the discussion of the proposed algorithm.